# Advancing nearshore and onshore tsunami hazard approximation with machine learning surrogates

Naveen Ragu Ramalingam[1], Kendra Johnson[2], Marco Pagani[2,3], and Mario L.V. Martina[1]

[1]University School for Advanced Studies - IUSS Pavia, 27100, Italy
[2]Global Earthquake Model Foundation, Pavia, Italy, 27100, Italy
[3]Institute of Catastrophe Risk Management, NTU, Singapore

**Correspondence:** Naveen Ragu Ramalingam (naveen.raguramalingam@iusspavia.it)

**Abstract.** Probabilistic tsunami hazard and risk assessment (PTHA and PTRA) are vital methodologies for computing tsunami risk and prompt measures to mitigate impacts. However, their application across extensive coastlines, spanning hundreds to thousands of kilometres, is limited by the computational costs of numerically intensive simulations. These simulations often require advanced computational resources like high-performance computing (HPC), and may yet necessitate reductions in resolution, fewer modelled scenarios, or use of simpler approximation schemes. To address these challenges, it is crucial to develop concepts and algorithms for reducing the number of events simulated and more efficiently approximate the needed simulation results. The case study presented herein, for a coastal region of Tohoku, Japan, utilises a limited number of tsunami simulations from submarine earthquakes along the subduction interface to build a wave propagation and inundation database. These simulation results are fit using a machine learning (ML) based variational encoder-decoder model. The ML model serves as a surrogate, predicting the tsunami waveform on the coast and the maximum inundation depths onshore at the different test sites. The performance of the surrogate models was assessed using a five-fold cross-validation assessment across the simulation events. Further, to understand their real world performance and generalisability, we benchmarked the ML surrogates against five distinct tsunami source models from the literature for historic events. Our results found the ML surrogate capable of approximating tsunami hazard on the coast and overland, using limited inputs at deep offshore locations and showcase their potential in efficient PTHA and PTRA.

## 1 Introduction

Tsunamis are potentially one of the most devastating natural hazards impacting life, property and the environment. More than 250,000 casualties and USD 280 billion in damage were caused by tsunami worldwide between 1998 and 2017 (Imamura et al., 2019), with the 2004 Indian Ocean and 2011 Tohoku tsunami events responsible for most of these losses. Tsunami hazard and risk assessments are fundamental in disaster management, as they facilitate the effective management of coastal regions and communities at risk of experiencing a tsunami(Mori et al., 2022). The results of tsunami hazard and risk assessment help plan and prioritise local and regional hazard mitigation efforts like land use and management, the engineering design of protective structures and buildings, tsunami monitoring and early warning systems, evacuation plans, and emergency response.

The simulation-based tsunami hazard analysis workflow consists of modelling different processes of the tsunami life cycle - generation, propagation and inundation (Behrens and Dias, 2015; Marras and Mandli, 2021) as depicted in Fig.1. Each of these tsunami processes requires forward numerical modelling at different spatial scales, with varying complexity and different numerical schemes. Many of these steps are computationally demanding and a substantial number of such simulations may be required in tsunami hazard analysis.

There are two broadly categorised approaches to tsunami hazard and risk assessment (Mori et al., 2018): deterministic and probabilistic approaches. The deterministic approach aims to study a limited number of large tsunami scenarios, such as historical events or the possible worst-case events. This approach has been most widely used as it requires fewer events to simulate and hence less computational effort. With only a single or limited number of tsunami scenarios modelled, we can estimate a tsunami hazard metric for the given scenarios, such as wave height at an offshore site, or inundation depth, run-up, etc. at an onshore location of interest for each scenario. Such results are easy to communicate and are useful in conservative decision-making activities such as evacuation planning. Instead, the probabilistic approach involves modelling a large number of possible tsunami events typically in the range of thousands(Gibbons et al., 2020) to million(Basili et al., 2021; Davies et al., 2018) to estimate the exceedance rate of the said tsunami hazard metric at a location or region of interest(Geist and Parsons, 2006; Grezio et al., 2017). This approach is complex and computationally demanding but allows the possibility of exploring different sources of uncertainty and making risk-informed decisions. When linked with fragility and loss models, probabilistic estimates of potential damage or loss of life and property are obtainable (Goda and De Risi, 2017).

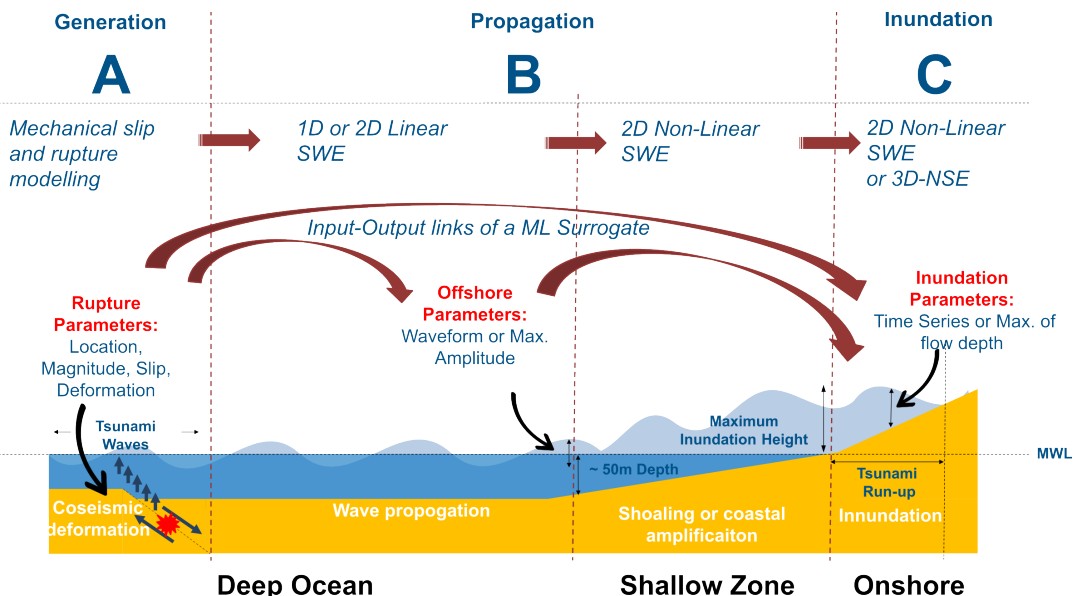

**Figure 1.** The sequence of a tsunami from the deep ocean generation and propagation, shallow coastal-zone shoaling, coastal inundation and interactions with the built environment, and the methods used to model each part of the sequence. The red dotted lines depict the interface between the tsunami forward modelling steps where different models can be coupled or linked.

The large computational burden from the many simulations needed in probabilistic tsunami modelling can limit their application, especially for onshore tsunami hazard and risk assessment where modelling of inundation processes is vital (Lorito et al., 2015; Grezio et al., 2017). Accurately simulating the tsunami wave shoaling process nearshore and the resulting inundation onshore necessitates the use of a 2D nonlinear shallow water equation (NLSWE) model at resolutions finer than at least 100 m on land. For scenarios where modelling turbulent tsunami forces onshore with precision is essential, a 3D Navier–Stokes Equation (NSE) model proves to be suitable (Marras and Mandli, 2021). Importantly, both the 2D NLSWE and 3D NSE models come with higher computational costs, exhibiting an exponential factor in comparison to the more computationally efficient 1D linear shallow water equation (LSWE) model commonly employed to model tsunami wave propagation in deeper oceanic regions. Figure 1 represents the off-to-on-shore tsunami forward modelling flow and the input-output links of the machine learning(ML) surrogates.

To handle or overcome this computational challenge of scale and the need to model a large number of events, various methods have been adopted (Behrens et al., 2021):

1. Reduce the number of events to be modelled, using sampling techniques, clustering, and other selection methods (Lorito et al., 2015; Williamson et al., 2020; Davies et al., 2022)

2. Approximations of results using methods like Green's Function, amplification factors, and reduced complexity models (Molinari et al., 2016; Løvholt et al., 2016; Glimsdal et al., 2019; Gailler et al., 2018; Grzan et al., 2021; Röbke et al., 2021)

3. Hardware and computational improvements like a nesting of grid domains, adaptive variable grid resolution, parallelisation and GPU-based acceleration, coupled multi-scale modelling, exascale codes (LeVeque et al., 2011; Shi et al., 2012; Oishi et al., 2015; Macías et al., 2017; Marras and Mandli, 2021; Folch et al., 2023)

4. Surrogates using statistical emulators and ML models (Sarri et al., 2012; Salmanidou et al., 2021; Mulia et al., 2018; Fauzi and Mizutani, 2019; Makinoshima et al., 2021; Fukutani et al., 2023; Mulia et al., 2022)

Running the numerical simulations, especially for modelling the tsunami inundation with an NLSWE model takes a lot of time and computational resources, for example accounting for about 13,600 GPU hours in the local PTHA study for Catania by Gibbons et al. (2020), refer Table 6 for runtimes of this study. Among the different approaches to reducing such computational and time burden, surrogates can provide an instantaneous approximation to the output of the numerical simulation. Surrogates are fit (trained) on a set of model inputs and outputs, called training data, to derive a mathematical or statistical relationship between the inputs and outputs; this provides a fast solution to otherwise time-consuming numerical simulations but may introduce some acceptable errors. Surrogates such as statistical emulators, can be fitted using a small training dataset built using a limited number of simulations. The application of such surrogates in tsunami modelling has been used for uncertainty quantification or sensitivity analysis (Sarri et al., 2012; de Baar and Roberts, 2017; Kotani et al., 2020; Salmanidou et al., 2021; Giles et al., 2021; Tozato et al., 2022) and hazard assessment (Fukutani et al., 2021, 2023; Lee et al., 2023), which are generally difficult to conduct in a brute-force approach where one would need to simulate all possible events explicitly or in

a real-time time setting where running high-fidelity models in limited time is difficult. Another type of surrogate used are the machine learning models (referred to as ML surrogates herein), which are also trained on the available input-output datasets using a supervised learning framework. Tsunami ML surrogates in comparison to statistical surrogate models, especially those based on deep neural networks, utilise a large training dataset to optimise the model parameters using backpropagation. This may require a much larger number of numerical simulations to create the relevant input-output datasets for training or fitting such a model. Using such a framework to prediction of a real-time tsunami is feasible and widely proposed for faster than real-time forecasting and early warning purposes (Mulia et al., 2018; Fauzi and Mizutani, 2019; Liu et al., 2021; Rodríguez et al., 2022; Kamiya et al., 2022; Mulia et al., 2022; Wang et al., 2023) especially with inputs derived directly from real-time sensors (Makinoshima et al., 2021; Mulia et al., 2022). See Table 1 for a comparison of different surrogate classes.

Models of varying complexity and resolution, as depicted in Fig.1 are often coupled to accurately simulate the generation and propagation of tsunamis from the deep ocean to the near shore and inundation onshore (Abrahams et al., 2023; Son et al., 2011). For probabilistic tsunami modelling with a large number of events, the typical approach is to use outputs of a given model level as boundary conditions to the next model in forward modelling flow, taking advantage of each model's varying complexity, resolution, and computational efficiency. In this study, a hybrid modelling framework is introduced (see Sect. 2). We suggest using a limited number of full simulations to build or train a tsunami ML surrogate and completing the most computationally demanding phases of tsunami forward modelling for the remaining events as direct final modelling using the ML surrogate. Figure 1 compares the different methods used in traditional tsunami forward modelling chains and as an alternative different connections for the setting up of a surrogate in the form of statistical emulators and ML models.

This framework recognises several key concepts, including reducing the number of events for numerical simulation, modular multi-scale modelling, and the use of ML-based surrogates for hazard approximation. By combining these ideas, the proposed framework provides a comprehensive approach to enhance the modelling process and efficiently achieve the final results. Many of the models and datasets needed in this framework are already available; we discuss how to put them together in a workflow and build the ML-based surrogate for the final tsunami modelling. Further, we conduct experiments to check the skill and usability of an ML surrogate for tsunami hazard approximation nearshore and onshore.

The current challenge lies in the need for an extensive size of the training dataset generally required by ML surrogates in tsunami modelling for training to generate accurate predictions. While they provide instantaneous results, the cost of developing the training data set for the ML surrogates may outweigh the benefit of instantaneous prediction. When ML models are not appropriately trained to understand the underlying physics or dynamics of the tsunami and use small training dataset, they may overfit and struggle to generalise well beyond the training data(out-of-training situations) (Seo, 2024).

To overcome this dependence on a large training size, our ML surrogate exploits the ability of the encoder-decoder network in dimensional reduction, feature representation, and sequence-to-sequence transformation for approximating tsunami hazard nearshore and onshore. We use tsunami data from a small set (about 500) of simple earthquake rupture scenarios which may provide a sufficient learning basis for the ML surrogate. We describe how a variational encoder-decoder (VED), a type of neural network model (see Sect.2.2.2), is trained to take the tsunami waveform at points where offshore depth is 100 m as input and predict the tsunami waveform at nearshore points with depths of 5 m and maximum inundation maps onshore for

**Table 1.** Comparison of recent work using surrogates for tsunami approximation

| Reference | Method | Simulation Size (no of events) | Input Parameter | Output Parameter |
|---|---|---|---|---|
| **Statistical Surrogate** | | | | |
| Salmanidou et al. (2021) | GPE | 60 | Seafloor displacement Parameters | Max height at coast |
| Tozato et al. (2022) | GPE, SVD | 50 | EQ Source Parameters (slip and rake) | Onshore max inundation depths, impact force |
| Fukutani et al. (2023) | GPE, SVD | 360 | EQ Magnitude ($M_w$), Seawall height | Onshore max inundation depths |
| Gopinathan et al. (2021) | GPE | 300 | EQ magnitude and location | Max height, velocity at coast |
| **ML Surrogate** | | | | |
| Liu et al. (2021) | CNN(VAE) | 1,300 | Short tsunami observation | Long tsunami forecast nearshore |
| Makinoshima et al. (2021) | CNN | 10,000 | Short tsunami observation | Onshore point inundation time series |
| Mulia et al. (2022) | Dense NN | 3,060 | Short tsunami observation(max) | Onshore max inundation depths |
| Rodríguez et al. (2022) | Dense NN | 16,000 | EQ Source Parameter | Max wave height, arrival time at coast |
| Núñez et al. (2022) | CNN | 6,776 | Offshore observation at 50m or 100m depth | Onshore point inundation time series |
| Cesario et al. (2023) | Regression Tree | 15,408 | EQ Source Parameter | Max wave height at a coast |
| de la Asunción (2024) | Dense NN | 128,000 | EQ Source Parameter | Alert level at coast |

**GPE**: Gaussian Process Emulator

**SVD**: Singular Value Decomposition

**CNN**: Convolutional Neural Network

**VAE**: Variational Autoencoder

**CNN**: Convolutional Neural Network

**Dense NN**: Dense Neural Network

three different locations along the coastal Tohoku region in Japan. Thus, skipping the computationally demanding modelling of non-linear processes in the shallow water regions nearshore and the inundation processes onshore for a given tsunami event. We check the generalisation ability of this model by testing the prediction error for a set of finite-fault rupture events of historic tsunami scenarios to evaluate the efficacy of this hybrid modelling approach.

## 2 A framework for approximating tsunami hazard nearshore and onshore at reduced computation cost

Our framework employs a hybrid modelling approach that integrates physics-based numerical simulation and data-driven ML models for tsunami hazard approximation. By combining the strengths of these models, we aim to represent the tsunami hazard for a coastal site of interest as the time series of the tsunami wave height near or along the shore and the max inundation depth onshore within a reasonable computational budget.

This section discusses the various components of the proposed framework as seen in Fig.2. The first step is the generation of synthetic data which is needed to train the ML surrogate model. This requires the full forward modelling of the tsunami generated from simple earthquake sources discussed in the design of events and recording the tsunami water level time series for such events at different depths, at 100 m depth and 5 m depth near shore, and the maximum inundation depth onshore for the event using an NLSWE tsunami model.

The second step is the development of ML models for use as surrogates, whose outputs are the near-shore tsunami height time series and the onshore maximum inundation depth. In the current study, the framework is applied and tested by comparing the prediction of the ML surrogate against results from numerical simulation for a portion of the synthetic events as hold-out testing and historic events for generalisation testing.

### 2.1 Generation of Synthetic Data

The characteristics of tsunamis in a region are intricately tied to factors such as earthquake sources and their recurrence, ocean bathymetry, shelf profile, coastal topography, and man-made structures like coastal defence and urban infrastructure. The rarity of tsunami events and the limited availability of comprehensive observational records across the coastal region make it impossible to construct a robust dataset for training and testing a ML model using purely historic observations.

Due to this data limitation, we instead attempt to create a diverse dataset for training and testing the ML surrogate to effectively capture the wide spectrum of tsunami dynamics in a region. The idea is to simulate earthquakes of different magnitude, different locations, with different slip amounts and rupture geometries generating tsunami waves of different amplitude and wavelengths offshore, causing inundation of varying patterns and extents onshore.

Thus, we model a wide variety of earthquake scenarios discussed in Sect.2.1.2 using a tsunami hydrodynamic model created for the Tohoku region (Sect.2.1.1) covering three test sites.

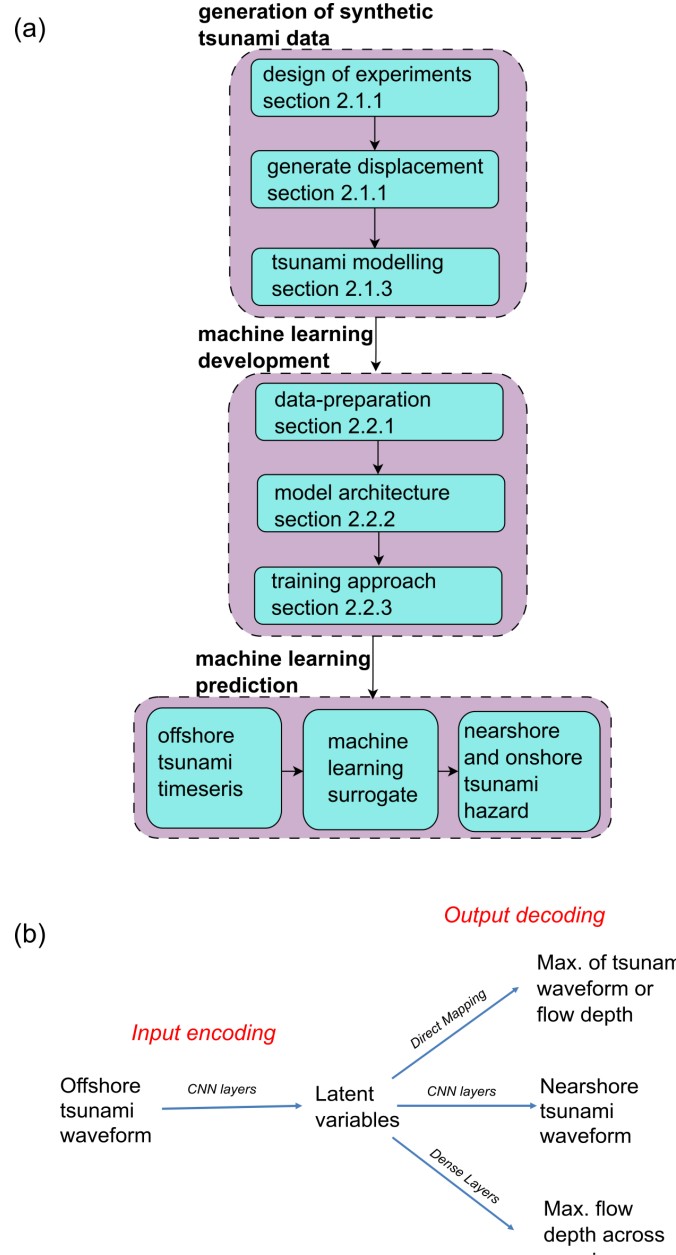

**Figure 2.** Overall framework for nearshore and onshore hazard approximation using an ML surrogate. (a) Flowchart of the procedure. (b) Different model structures based on prediction types.

### 2.1.1 Tsunami model and test locations

Three locations with different coastal configurations are identified for this study – Rikuzentakata (enclosed bay), Ishinomaki (shielded), and Sendai (open bay), – where different coastal processes like shoaling, refraction, reflection, and resonance are expected to impact the tsunami nearshore results and provide varied settings for testing the proposed methodology of developing an ML-based surrogate.

For the simulation of the synthetic events, and recording their offshore waveform (water level time series) and onshore flood inundation depths, a tsunami model based on GeoClaw Version 5.7.1, (Clawpack Development Team, 2020) was developed. This GeoClaw model covers the Pacific Side of the Honshu island and solves the depth-averaged NLSWE using adaptive mesh refinement using rectangular grids in the geographic coordinate system (latitude and longitude) for the base Level 0 at resolution 0.01215 degrees.

The adaptive mesh refinement ranges from level 1 of resolution 0.006075 degrees for the overall model domain, Level 2 of resolution 0.00405 degrees at bathymetric depths around 100 m to 850 m and level 3 of resolution 0.00135 degrees for bathymetric depths less than 100 m as in Fig.3(a). An additional refinement is enforced for the three test sites resulting in the finest grids of 0.00045 degrees roughly equivalent to 50 m in resolution for capturing the inundation onshore as seen in Fig.3(b).

The model uses topographic data from the Japan cabinet project available in 1350-450-150-50m resolution, the final 50 m resolution grids of this dataset were superimposed with COP-DEM, a Digital Surface Model (DSM) representing the elevation of the Earth's surface including buildings, infrastructure and vegetation with native resolution of 30 m, and a coastal tsunami defence elevation dataset also available from the cabinet project for representing the onshore elevation more realistically for the tsunami inundation modelling.

The virtual gauges are set at depths of 5 m and 100 m as shown in Fig.3(b) to record the elevation of water level at regular time intervals and fixed grid monitoring is used to record the maxima of the inundation depths from each event for the three coastal regions for each tsunami simulation. Each tsunami simulation is run for a 6-hour duration from the onset of the tsunami. Tides and wave components are ignored and the initial water level condition is set at zero mean sea level to consider a still ocean condition.

The model was tested and calibrated using the 2011 Tohoku historical event's gauge and inundation survey data, see results available in Supplements Fig.S1 and Fig.S2.

### 2.1.2 Design of experiments(DOE)

The design of the experiment consists of a total of 559 hypothetical events of two categories of rupture, distinguished by their geometry and slip distribution: (a) Type A - ruptures represented by a single rectangular planar surface with homogeneous slip and (b) Type B - ruptures that combine numerous smaller rectangular planar surfaces (i.e., sub-ruptures), each of which has homogeneous slip, such that the rupture surface can bend and the slip distribution can be heterogeneous. The number of events in the DOE is constrained by available computational resources and our goal to use a feasible number of training events and

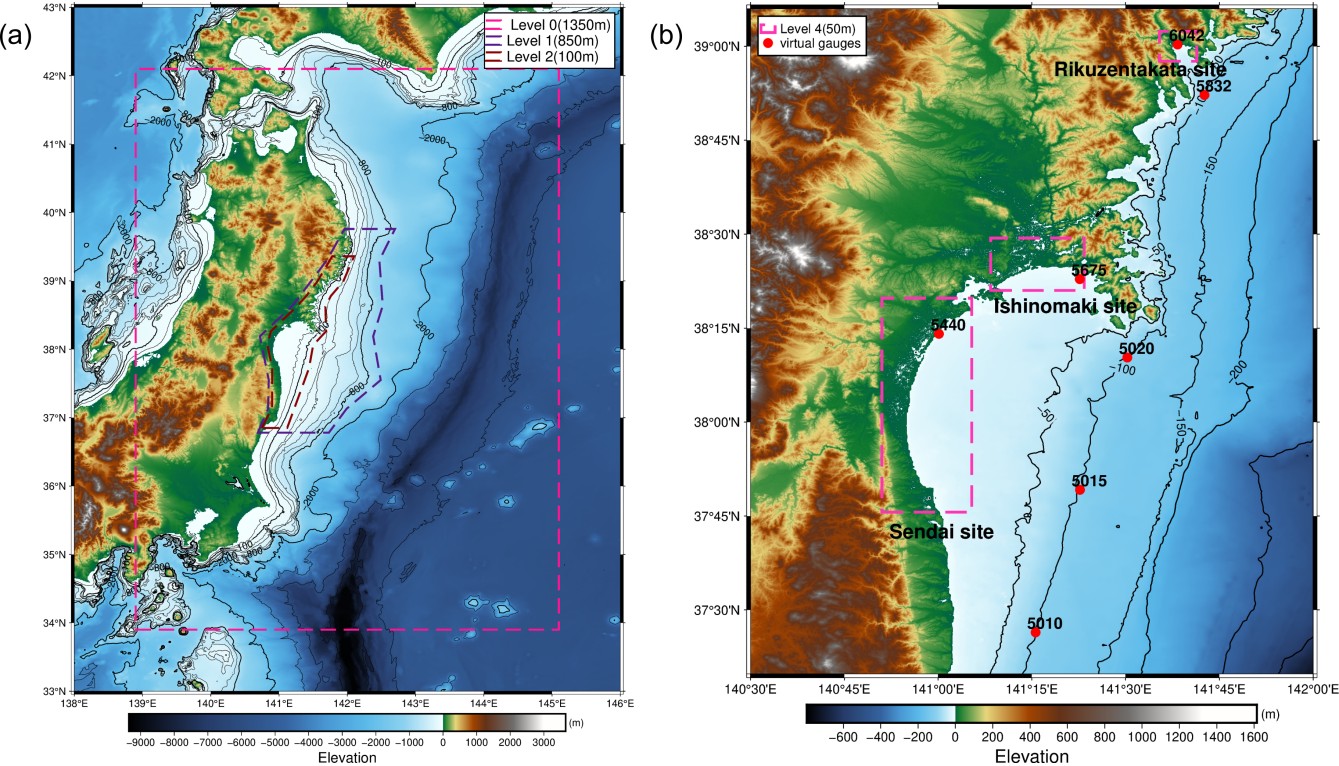

**Figure 3.** (a)Tsunami model coverage and adaptive grid resolution system. (b)Virtual Gauges and inundation sites with high resolution nested domains.

maximise the efficiency of the surrogate model. The source scenarios are modelled by adapting procedures previously applied by Gusman et al. (2014) and Mulia et al. (2018).

A total of 119 locations were selected as the top centre of the faults for modelling hypothetical tsunamigenic earthquakes of Type A. These events span $M_w$ 7.5 - 9.0 at an interval of 0.5 and are uniformly distributed over the Tohoku subduction

interface, see Fig.4(a) and 4(b). This results in a potential 476 events (119 locations x 4 magnitudes). The $M_w$ 9.0 events are restricted to locations where the centre of the rupture's top edge is shallower than 16 km. Deeper events cause unrealistic uplift on large inland portions of the study region and are unlikely to cause tsunami. To ensure realistic modelling and prohibit these events from adversely affecting the quality of the surrogate training, these events were excluded, leaving 383 Type A events.

Multi-fault ruptures of Type B were created using a combination of 6 to 12 planar sub-faults similar to the unit sources used

in NOAA SIFT database (Gica et al., 2008) of length 100 km and width 50 km are created as in Fig.4(c). The event magnitudes range $M_w$ 8.68 - 9.08, and the ruptures are distributed along the shallow section of the Tohoku subduction interface. The bottom centre of the rupture edges are at depths between 17-28 km. The slip distributions are modelled as a skewed normal distribution where the average combined slip value is between 10 and 20 m. The scenarios varied by the number of faults involved: scenarios with 6 faults were assigned a slip of 10 m, scenarios with 8 faults had slips of 10 and 15 m, 10 faults with

15 m, and 12 faults with slips of 15 and 20 m. This systematic variation led to a total of 176 different Type B earthquake scenarios.

Information on depth, slip, strike, and dip (see Table 2) is derived from the Slab2 model of the Japan trench (Hayes et al., 2018), and the rake is always set at 90 degrees (Aki and Richards, 2002). The seafloor deformation is analytically modelled assuming homogeneous slip for the rupture or sub-ruptures using Okada solution (Okada, 1985) with the value of rupture

length, width, and slip scaled (see Table 2) based on the magnitude of the event (Strasser et al., 2010). We consider that the co-seismic displacement is instantaneous and equivalent to the sea surface displacement generating the tsunami. This initial sea surface displacement is modelled to match the base resolution of the tsunami model at 0.01215 degrees grids.

In summary, the DOE for training the surrogate model considered two main factors: (A) moment magnitude, which determines the profile of displacement (length, width, and slip amount) based on the moment magnitude-area scaling relationship

(Strasser et al., 2010), and (B) the location of the events where fault parameters such as depth, dip, rake, and strike are derived from the Slab2 dataset.

**Table 2.** Earthquake rupture parameters for Type A and Type B, the rake value is always 90 degrees.

| Type | $M_w$ | Length (km) | Width (km) | Displ. (m) | Depth (km) | Dip (degrees) | Strike (degrees) |
|---|---|---|---|---|---|---|---|
| **Type A Min** | 7.5 | 81.37 | 56.29 | 0.24 | 10.2 | 5.54 | 187.20 |
| **Type A Max** | 9 | 613.76 | 189.23 | 3.30 | 45.7 | 17 | 225.78 |
| **Type B Min** | 8.68 | 300 | 100 | 4.72 | 17.01 | 8.37 | 188.72 |
| **Type B Max** | 9.08 | 600 | 100 | 17.36 | 28.98 | 16.53 | 222.27 |

### 2.1.3 Test Events

Along with using a random subset of the events from the design of experiments, we additionally model a set of 5 events to evaluate the performance of the ML model. This is to test the model over a generalised dataset which is different from the

200 training dataset used in building the model. The numerically simulated tsunami are known to be sensitive to the earthquake slip, fault geometry, rupture mechanisms, and the discretisation used (Gibbons et al., 2022; Goda et al., 2014) and their resulting wave profile, direction, and inundation may vary significantly. Although there is significant variation in the characteristics of the different tsunami events generated in the design of experiments to train and test the ML surrogate, we add these additional historic test events with complex heterogeneous slip, events with instantaneous and time dependent slip displacement and

205 events located beyond the Tohoku source region used for training, as seen in Fig.4. These include the following events described below.

1. **2011 Tohoku Earthquake**:

   – Test A(instantaneous displacement) using Fujii et al. (2011)

   – Test E(time-dependent displacement) using Yamazaki et al. (2018)

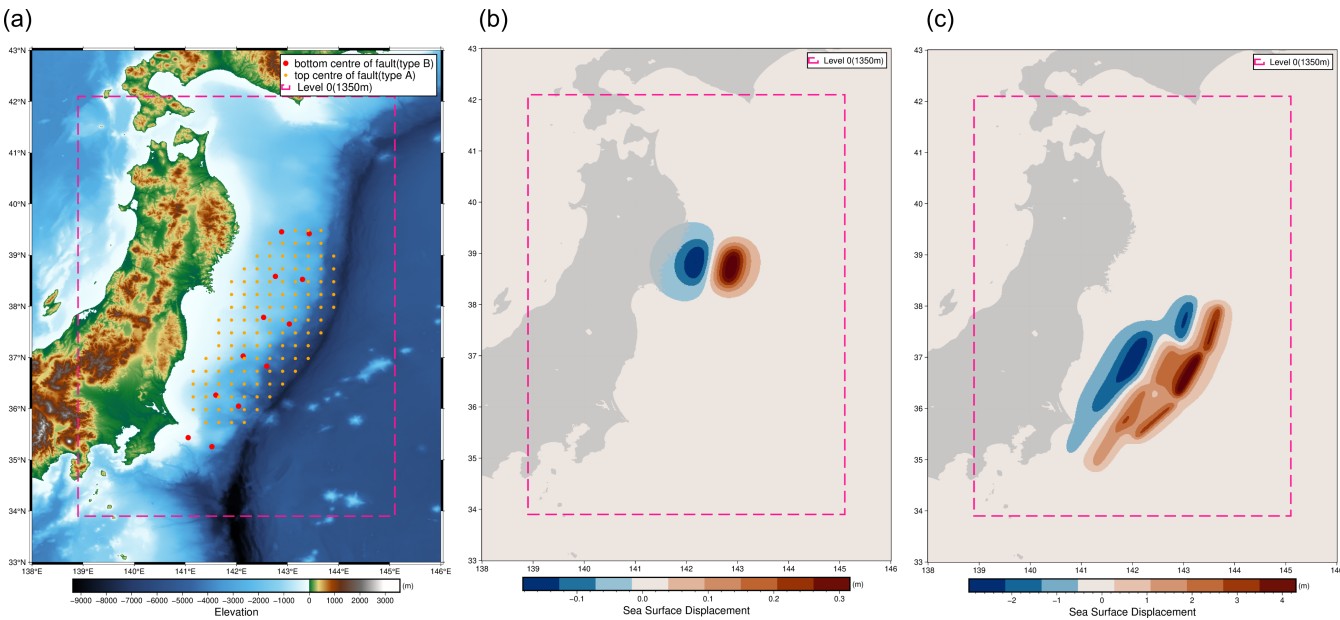

**Figure 4.** Example initial displacement for the two rupture types used in the design of experiment for the tsunami dataset. (a) Fault Location for type A and B, (b) Type A - Event 35, (c) Type B - Event 33

2. **1933 Sanriku Earthquake**:

   – Test B(outer rise event with normal faulting) using Okal et al. (2016)

3. **1896 Sanriku Earthquake**:

   – Test C(outside Tohoku source region considered in DOE) using Satake et al. (2017)

4. **1968 Tokachi-Oki Earthquake**:

   – Test D(outside Tohoku source region considered in DOE) using Riko et al. (2001)

## 2.2   Machine Learning Model

Previous works (Fauzi and Mizutani, 2019; Liu et al., 2021; Makinoshima et al., 2021; Núñez et al., 2022; Mulia et al., 2022; Rim et al., 2022) that focused on using ML models for use in tsunami forecasting and early warning needs have also used neural networks in the form of CNN and MLP models. These models are usually trained to take short duration (20 - 30 min) inputs from sensors like offshore ocean bottom pressure sensors and geodetic sensors (GNSS) for their tsunami hazard prediction. This is due to the constraints of a short lead time in the event of a tsunami. Furthermore, the tsunami observation network is designed with a given earthquake source region in focus, leading to the design and training of the ML model which is constrained to a very specific regional and source setting.

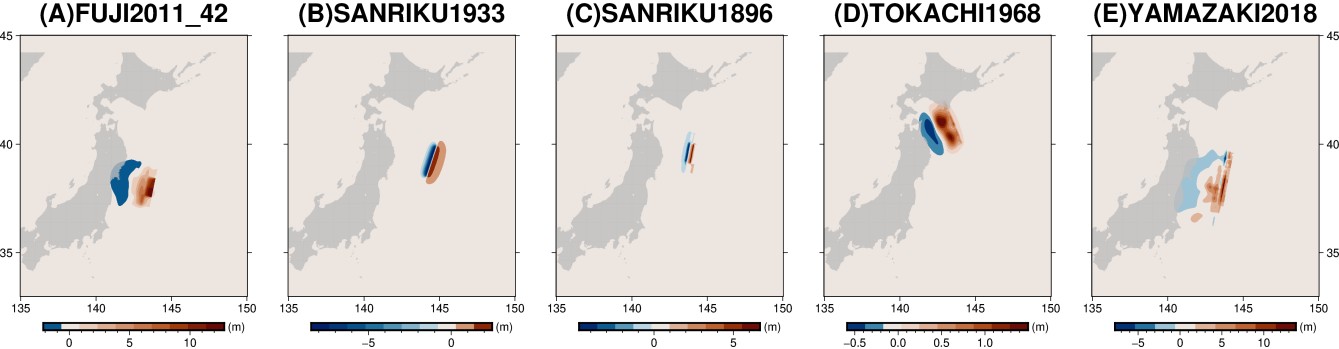

**Figure 5.** Initial sea surface displacement for the 5 test events.

In contrast, we discuss two ML surrogate models in this work with the main difference being the output of the model: either a time series of the water surface elevation or the maximum inundation depth across a set of onshore locations. The design of the ML model or surrogate for the current framework focuses on:

1. The model architecture, that should be able to train and fit with a relatively small number of events such that it can serve the purpose of solving the computational constraint for its use as a surrogate model.

2. The model has a balanced performance, it can predict the hazard sufficiently well for small and large-magnitude events across the domain of interest.

3. The model training is not sensitive to training datasets, that is, it should not overfit the data and should be capable of predicting results for different ruptures as long as the offshore water level amplitude is available.

4. The model design should be able to connect easily with available outputs from other regional propagation models and be easily replicated across different coastal configurations.

### 2.2.1 Data preparation

The ML model in this study uses two types of datasets from the tsunami simulations: (a) tsunami waveforms - water level time series and (b) maximum inundation depth map. Figure 6 provides a summary of the dataset for the three test locations - inundation depths, area and maximum amplitude at the virtual gauges.

For each of the events, we process the simulated water level time series at the virtual gauges and the maximum inundation map in the following steps. All events where the tsunami water level does not cross a set threshold of 0.1 m at the selected deep offshore virtual gauges of 100 m depth and 0.5 m at the shallow nearshore virtual gauge of 5 m depth are ignored as they result in negligible tsunami inundation.

At the instance of a time when this threshold is crossed at the shallow gauge, a simulation time window of 240 minutes is selected to calculate a uniformly sampled wave amplitude time series with 1024 data points. As many of these local source

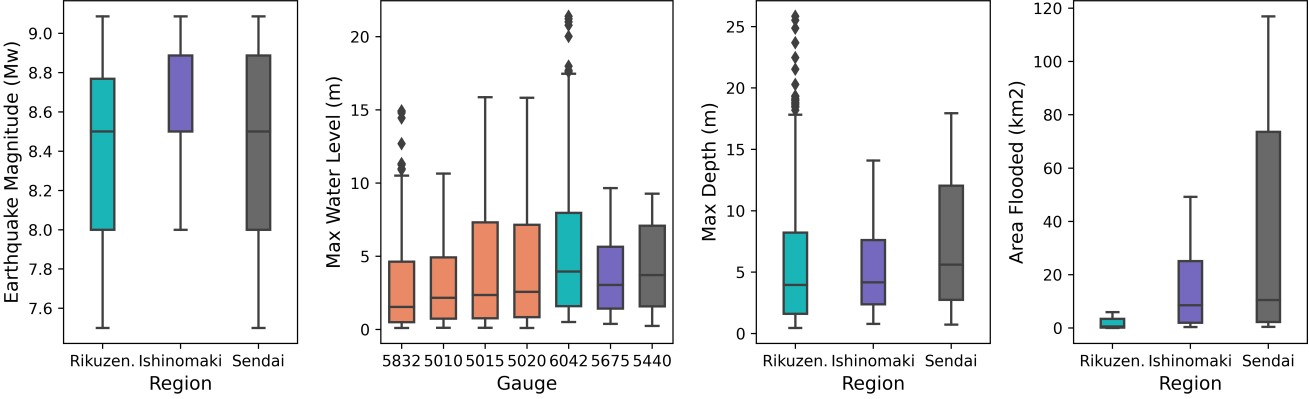

**Figure 6.** Box plot showing variability in the offshore and nearshore gauges and the inundation properties for the three test sites.

events cause significant local deformation to the bathymetry , which is captured in the and water level time series. This offset is removed from the time series data at both nearshore and offshore virtual gauges (e.g. Fig.7) as a preprocessing step for reducing the complexity in the dataset which can help in training the model.

In the case of the maximum inundation maps, all grid points that are flooded across the entire simulation dataset of all the modelled events are selected as a fixed set of prediction locations shown in Fig.8 for Rikuzentakata, and the maximum inundation for each event at each such location is stored as a 1D array. This yields 6648 grid points for the Rikuzentakata area, 54671 grid points for the Ishinomaki area, and 129941 points for the Sendai area.

### 2.2.2 Variational Encoder-Decoder(VED) - Model architecture and training approach

Our design of the "VED" architecture extends upon existing statistical and ML models used for tsunami forecasting and rapid modelling. Mulia et al. (2018) used the feature space from the principal component analysis (PCA) as a form of dimensional reduction to search the closest inundation map from a large simulation database against results from a quick low-resolution simulation and interpolate new high-resolution results using these samples. ML based models that are trained on tsunami simulation databases, in particular convolutional neural networks (CNN), have also been used to predict the full-time series of the water level elevation using sparse observational inputs (Liu et al., 2021; Makinoshima et al., 2021; Núñez et al., 2022). In the case of prediction for maximum inundation depth maps, a fully connected or dense layers have been used to efficiently map the output across the large set of locations as in Fauzi and Mizutani (2019); Mulia et al. (2020, 2022).

The deep neural network models used in the above work for tsunami predictions typically need a large number of training examples to learn a useful representation and avoid overfitting the parameters of their different layers. Certain model architectures and training schemes can help them become more learning efficient, generalise better, and provide higher prediction performance. One such approach is encoding-decoding, a class of supervised ML aimed at training a neural network to learn a

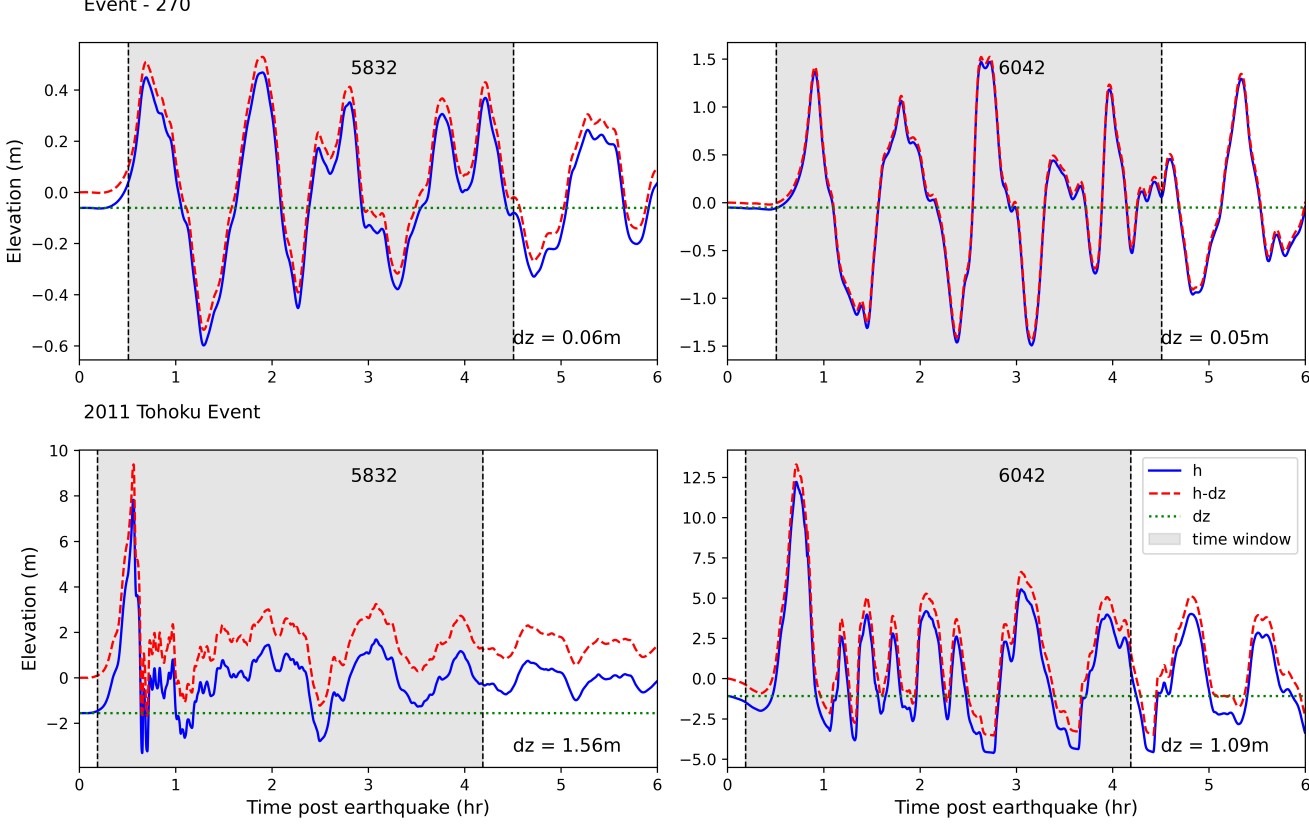

**Figure 7.** Correction of the waveform by removing deformation, with a 4-hour selection window based on an offshore gauge crossing threshold of 10 cm.

lower-dimensional representation of the input that can be used to construct the output. It consists of three parts: encoder, latent variables, and decoder.

$$z = f(w_f, b_f, x) \tag{1}$$

$$y = g(w_g, b_g, z) \tag{2}$$

The encoder function $f(.)$ with its learnable parameters, weights $w_f$ and biases $b_f$ maps the input data $x$ to a reduced number of latent variables $z$ and a decoder function $g(.)$ with its learnable parameters, weights $w_g$ and biases $b_g$ maps the latent variables $z$ back to the high dimensional data space as the construction of the output $y$ see Eq.1 and 2. The encoder

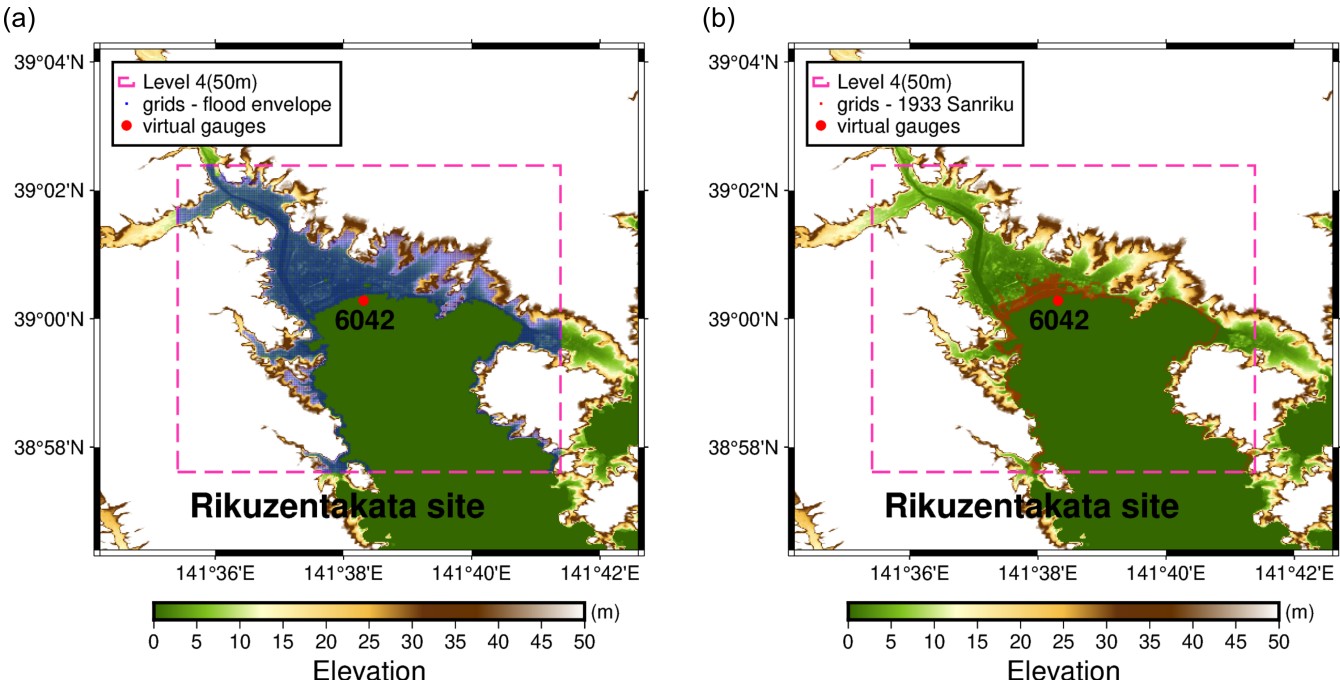

**Figure 8.** Location of grids affected by inundation: (a) within the overall flood envelope from all DOE simulations and (b) for a single 1933 Sanriku test event.

- decoder(ED) is trained to minimise the reconstruction error between the input data and the output of the decoder network using back propagation optimisation. Similar to Liu et al. (2021) we use convolutional layers and variational encoding; see Fig.9. The encoding convolutional layers and max pool operations of the encoders perform dimensionality reduction on the time series data at the input gauges similar to a principle component analysis (PCA) of Mulia et al. (2018) and in inverse the decoding transposed convolutional or dense layers performs the necessary transformation to predict the output time series or multi-location inundation depths, see Tables S1 and S2.

Variational encoding maps the input data to the probabilistic distribution of the latent variables $z$, this makes VEDs more flexible in capturing the underlying data distribution and modelling the epistemic uncertainty in the input encoding process. Compared to a deterministic encoders, the latent space is mapped as continuous and smooth variables, making for a more interpretable and structured representation of the data in the encoding latent variables Liu et al. (2021).

We use two VED models as the ML-based surrogates, to predict the water level time series(waveform) at a gauge location and the flood inundation footprint in the form of the maximum inundation depth. Early in the study we evaluated the prediction of two inundation parameters - maximum inundation height and maximum inundation depth, with the model predicting the maximum inundation depth better. We do not discuss the result for the maximum inundation height in the study. The models receives its input $x$ in the form of a stack of time series from the selected offshore gauge/s, which is encoded into the latent variables $z$ by four convolutional layers with kernel size 3 and padding 1 using a Leaky ReLU (0.5) activation which helps

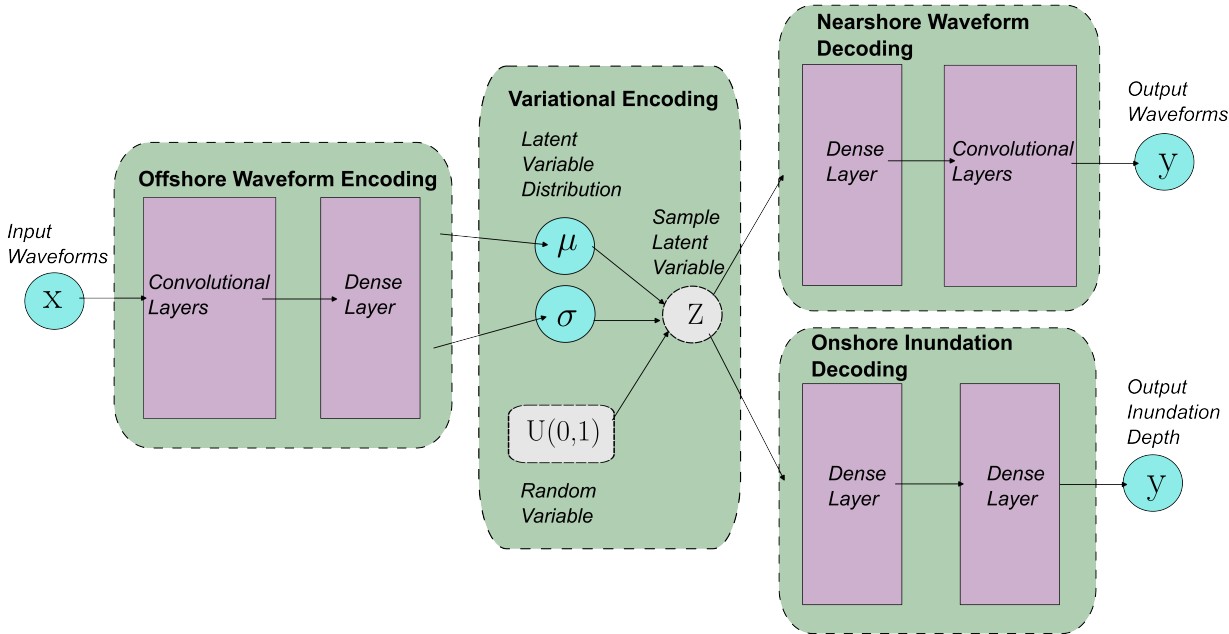

**Figure 9.** Schematic of a Variational Encoder-Decoder.

introduce non-linearity and prevent vanishing gradient during training. After each convolutional layer, we perform a max-pooling operation to reduce spatial dimensions on the feature maps. The choice of the number of layers, channels, kernel size and the input data sampling rate are interdependent and were decided by testing multiple configuration, Tables S1 and S2 describes the two surrogate architectures used in the study. For the encoding in the onshore VED model, we additionally apply batch normalisation after the first max pooling layer to help stabilise the training as we use a single batch training and a dropout layer with a rate of 0.1 is applied after the final max pooling layer as additional regularisation, refer other operations in S2. The latent variables $z$ are encoded as the mean and sigma values of a Gaussian distribution using a reparametrisation method (Kingma and Welling, 2014).

The decoding layers take the sample from the latent space and reconstruct the outputs. For near-shore VED, to predict the nearshore time series of the water level, we adapt the encoder architecture in reverse using four transposed convolutional layers. For the onshore VED, to predict the max inundation depths we use a dense layer that maps the sample from the latent space to the outputs i.e. the max inundation depth directly. The model hyperparameters including the number of latent variables, a learning rate of 0.0005, and no weight decay, were chosen through rough experimentation to achieve optimal results. Table 3

summarises the size of input, output, and latent variables used in the two surrogate models. Supplements Table.S1 and Table.S2 provide more information on the model layers and operations used in the two surrogate models.

The loss function used to train the models comprises two components: the prediction loss and the Kullback-Leibler (KL) divergence loss. The prediction loss measures the difference between the constructed data to the output data in terms of the mean squared error. The KL divergence loss (Kingma and Welling, 2014; Liu et al., 2021) encourages the latent space distribution to be close to a unit Gaussian distribution, acting as a regularisation term. This additional regularisation also helps prevent overfitting and improve generalisation. The components of the loss function in Equation 3 can also be weighted to improve the overall model prediction.

$$\text{Total Loss} = \underbrace{\frac{1}{2}\sum_{i=1}^{n}(y_i - \hat{y}_i)^2}_{\text{MSE (Prediction Loss)}} + \underbrace{\text{KL}(q(z|x)||p(z))}_{\text{KL Divergence Loss}} \tag{3}$$

Given the relatively small size of our training dataset, we employ K-fold cross-validation with five folds to fully use the available data set for training and testing. The data set is randomly partitioned into five equal-sized folds. Each fold is used once as the test set while the remaining four folds are used for model training. This process is repeated five times, with each fold serving as the test set exactly once. The characteristics and parameter ranges for each test fold as when all the test folds are combined together mirror that of the overall training dataset as found in Table 2 and Figure 6. This helps in assessing the model's generalisation ability and sensitivity to overfit with such limited training information (Mulia et al., 2020). We conducted the training in a single batch. This approach differs from conventional training settings, where mini-batch training is adopted to handle large training datasets. The single batch approach allows the model parameters to be optimised using the entire fold's dataset in each training iteration, effectively leveraging all available information to update its parameters. This strategy of data splitting, cross validation, and VED model architecture facilitates better convergence and helps prevent the model overfit during training or data leakage during evaluation.

Applying an ensemble approach to the prediction, we use the variational encoding 9 to generate 100 sample predictions for each test event of the fold. Further, when generating prediction for the historic event in 2.1.3 we use all the five models trained as part of the 5-fold cross validation to generate 5 x 100 predictions, thus using the overall dataset for model training and prediction for these test events.

## 3 Metrics for evaluation

To assess the general fit between the numerical simulation and ML predicted values, the mean squared error (MSE), the coefficient of determination ($R^2$), and the goodness of fit ($G$) are used. MSE quantifies the average squared difference between simulated and predicted values, with lower values indicating better model performance. $R^2$ measures the proportion of variance in the dependent variable (simulations) that is predictable from the independent variable (predictions), providing information on the accuracy of the model based on the correlation or dependence between simulations and prediction, with high values

**Table 3.** Hyperparameters for Onshore and Nearshore Surrogates in Different Regions.

|  | Region | Number of Latent Variables | Input Dimensions | Output Dimensions | Training Size |
|---|---|---|---|---|---|
| **Nearshore Surrogate** | Rikuzentakata | 25 | 1 x 1024 | 1 x 1024 | 490 |
|  | Ishinomaki | 150 | 3 x 1024 | 1 x 1024 | 425 |
|  | Sendai | 100 | 3 x 1024 | 1 x 1024 | 465 |
| **Onshore Surrogate** | Rikuzentakata | 10 | 1 x 1024 | 6648 | 490 |
|  | Ishinomaki | 50 | 3 x 1024 | 54671 | 425 |
|  | Sendai | 30 | 3 x 1024 | 129941 | 465 |

**Table 4.** Summary of Evaluation Metrics and Formulas

| Metric | Formula | Focus |
|---|---|---|
| Mean Squared Error (MSE) | $\frac{1}{n}\sum_{i=1}^{n}(y_i - \hat{y}_i)^2$ | Mean Error |
| Coefficient of Determination ($R^2$) | $1 - \frac{\sum_{i=1}^{n}(y_i - \hat{y}_i)^2}{\sum_{i=1}^{n}(y_i - \bar{y})^2}$ | Correlation or dependence |
| Goodness of fit ($G$) | $1 - \frac{2\sum_{i=1}^{n}(y_i \cdot \hat{y}_i)}{\sum_{i=1}^{n}y_i^2 + \sum_{i=1}^{n}\hat{y}_i^2}$ | General Performance |
| Relative Peak Deviation (RPD) | $\frac{y_{\text{peak}} - \hat{y}_{\text{peak}}}{y_{\text{peak}}}$ | Delay in peak |
| Maximum Peak Delay (MPD) | $t_{y_{\text{peak,max}}} - t_{\hat{y}_{\text{peak,max}}}$ | Peak of the waveform |
| Accuracy Score (A) | $\frac{\text{True Predictions}}{\text{Number of locations}}$ | Flooded Area/Mapping |
| L2Norm | $\sqrt{\sum_{i=1}^{n}(y_i - \hat{y}_i)^2}$ | Magnitude of event |

close to 1 indicating a good result. $G$ is used as a cost function to measure the disagreement between observed and predicted values, considering both the magnitude and direction of the deviations, with a lower value close to 0 being better. Lastly, we used the L2Norm to estimate the magnitude of the event, using the vector representing the time series at the test gauge or the maximum inundation depths for a region.

In the case of the nearshore surrogate, where accurately predicting the peaks and their timing is crucial, Relative Peak Deviation (RPD) and Maximum Peak Delay (MPD) are used. RPD measures the relative difference between simulated and predicted peak values, providing insight into peak accuracy. MPD evaluates the time delay between highest simulated and predicted peaks, crucial for assessing the timing accuracy of peak. For the onshore surrogates, where accurately predicting inundation extents and depth is the focus, the accuracy score quantifies the proportion of correctly predicted inundation locations relative to the total number of locations. This metric provides a clear indication of the model's predictive performance in accurately mapping inundation for depths above the threshold of 10 cm or a select depth class (say 0.1 m to 1 m). These metrics collectively offer a comprehensive evaluation of the ML surrogate's prediction.

## 4 Results

### 4.1 General Fit

As a first step, we evaluate the sensitivity of the learning or parameter optimisation to the training data as part of the five-fold cross-validation study. The mean square error values are summarised in Table 5 for each of the folds. For the nearshore surrogate (predicts the time series) it ranges between 0.042 - 0.197 and for the onshore surrogate (predicts the max inundation depth map) it ranges between 0.169 - 1.129 at the three test sites. For the nearshore surrogate, the relatively consistent MSE across folds suggests that the model is robust, with no significant overfitting and an overall good fit to the data. However, for the onshore surrogate, there is noticeable sensitivity to the training data, as evidenced by slightly higher MSE values for certain locations, such as Rikuzentakata (fold 2) and Ishinomaki (fold 0).

This increased sensitivity could be attributed to two factors, namely the training size and the complexity of the output. The smaller size of the training set may lead to higher variance in model performance, as the model has less data to learn from, making it more sensitive to the specific events included in each training fold. The onshore surrogate is tasked with predicting inundation, which is inherently more complex and variable compared to nearshore waveforms. This complexity can further lead to greater variation in model performance across different folds, especially with limited training data. In summary, the observed variance in MSE across folds is not random but is influenced by the size and complexity of the training data. For the nearshore surrogate, the model demonstrates stable performance, while the onshore surrogate shows some sensitivity, which is expected given the challenging nature of the inundation predictions.

We also record the maximum elevation values from both the surrogate for each test event in Sect.2.1.2 and compare it with the simulation values of the GeoClaw tsunami model described in Sect.2.1.1 using the scatter plots of Figures 10 and 11. The predictions are in good agreement, well correlated with the simulation values and lay close to the true line. In case of the nearshore surrogates, we notice larger uncertainty for Sendai compared to the the other test sites, while for the onshore surrogates, Ishinomaki has more uncertainty. The plot also marks the prediction for the five historic test events, discussed further in Sect.4.4. The prediction of the maximum elevation for the waveforms and the maxima of the maximum inundation depth show good fit with little underestimation or overestimation (Figures 10 and 11) and highlight the generalisation of the model across unseen data.

### 4.2 Nearshore Approximation

The nearshore surrogate predicts waveforms of 4 hr duration; the time series predictions for gauge 6042 at Rikuzentakata, gauge 5675 at Ishinomaki, gauge 5440 at Sendai for five random events from the fold 0 test set are shown in Fig.12. These figures show the mean prediction along with the 2 standard deviation uncertainty bands for all the three gauges. This uncertainty represents the epistemic uncertainty of the ML model from a variational latent space. They show good agreement with the simulation, but the simulation values does not always lie within the uncertainty band of the prediction, as seen in event 319 at Rikuzentakata for the full time period. Examples such as Type B (Event 14) that show an excellent match and a narrow uncertainty range can be linked to training events sharing the same fault location but different slip distributions for the test

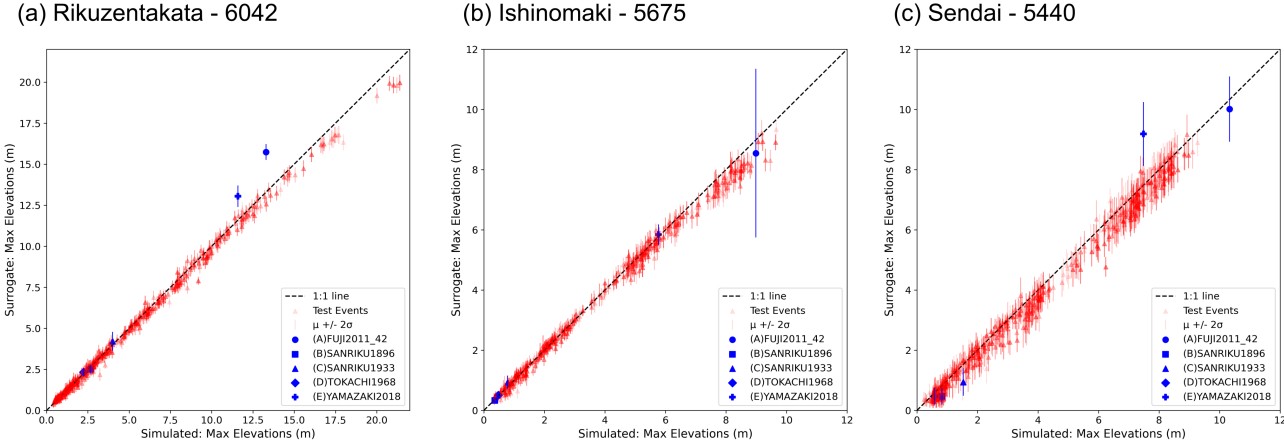

**Figure 10.** Prediction of maximum water levels from the nearshore surrogate for test and historic events.

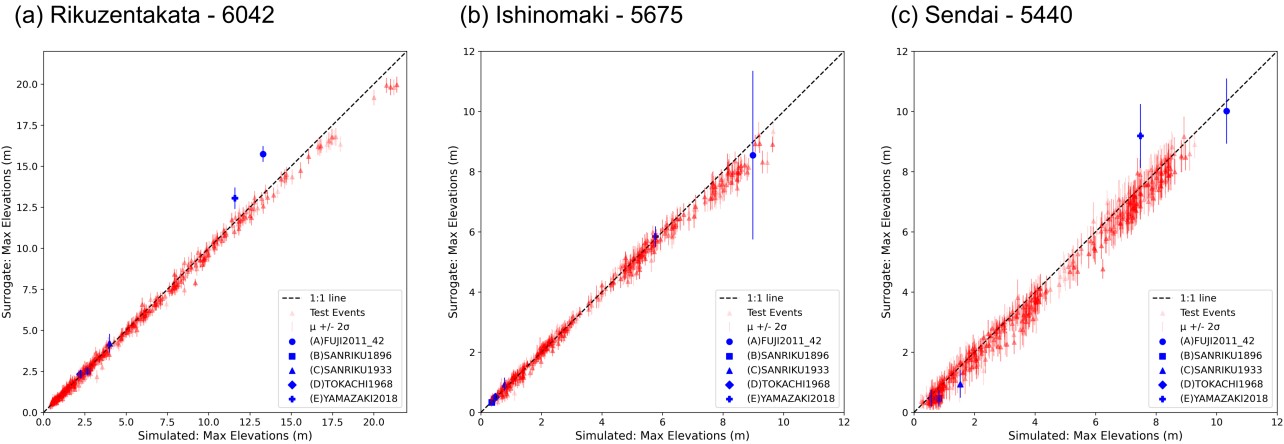

**Figure 11.** Prediction of maximum inundation depth from the onshore surrogate for test and historic events.

**Table 5.** Results of the k-fold cross validation using means square error loss metric for the two types of surrogate models tested at the three test sites.

| | Region | MSE | | | | |
|---|---|---|---|---|---|---|
| | | Fold 0 | Fold 1 | Fold 2 | Fold 3 | Fold 4 |
| **Nearshore Surrogate** | Rikuzentakata | 0.098 | 0.057 | 0.042 | 0.122 | 0.111 |
| | Ishinomaki | 0.042 | 0.065 | 0.061 | 0.071 | 0.053 |
| | Sendai | 0.197 | 0.087 | 0.067 | 0.168 | 0.059 |
| **Onshore Surrogate** | Rikuzentakata | 0.179 | 0.406 | 1.129 | 0.262 | 0.169 |
| | Ishinomaki | 0.886 | 0.349 | 0.327 | 0.282 | 0.373 |
| | Sendai | 0.35 | 0.263 | 0.377 | 0.261 | 0.576 |

event. Furthermore, the uncertainty range provides a useful indicator of the stability and sensitivity of the selection of model parameters and latent variables. We also evaluate the performance of the mean prediction using the evaluation metrics for the time series in Table 4. There is a relative peak deviation of up to 0.2 and a maximum peak delay of up to 2 hours when the magnitude of the highest peak is not captured accurately.

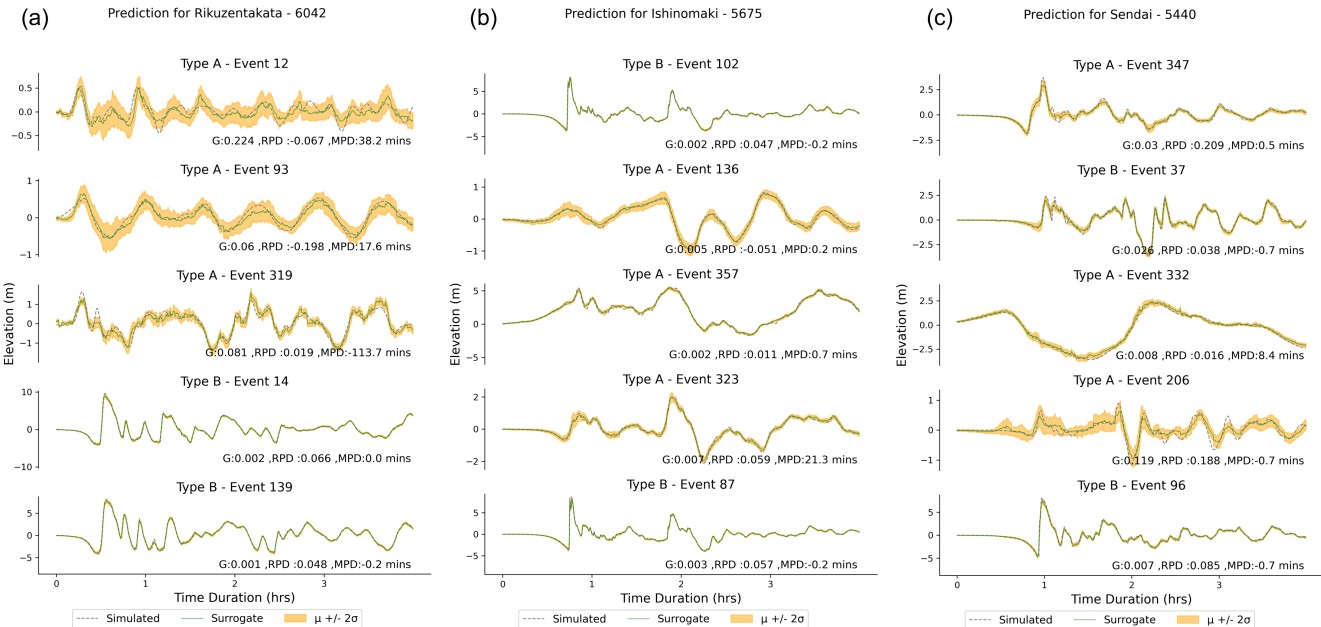

**Figure 12.** Prediction examples from the nearshore surrogate for the DOE test events.

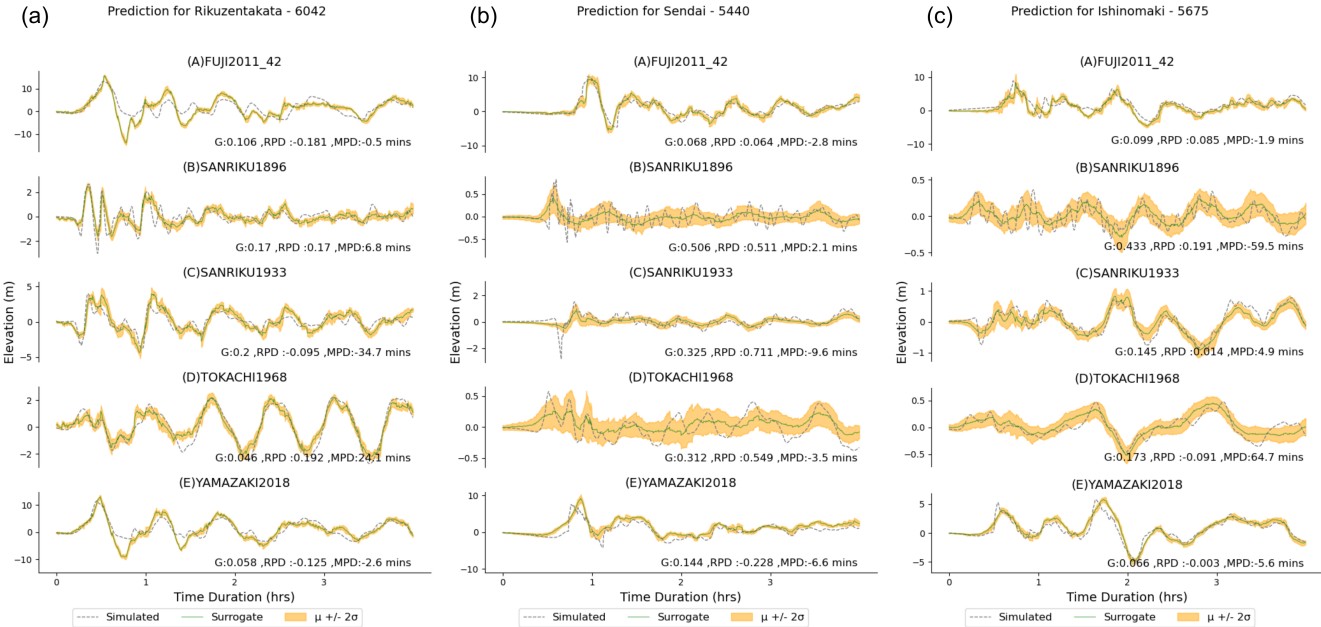

**Figure 13.** Predictions from the nearshore surrogate for the historical events.

## 4.3 Onshore Approximation

With the onshore surrogate we predict the maximum inundation depth at the selected grid location for the three test sites. The prediction maps at Rikuzentakata for five events from the fold 0 test set are shown in Fig.14. We plot the true GeoClaw simulation values and the mean predictions of the surrogate in columns 1 and 2. The misfit between simulation with the mean ML prediction and +/- 2 standard deviation values are plotted in columns 3-5. The uncertainty in the onshore surrogate leads to variability in both the mapped inundation area and the water depth at each grid location for the predictions. This uncertainty depends on the ML parameters and the latent variables. The predictions show good agreement with the simulations, though there is a tendency of the having some artefact flooding, i.e. to predict inundation at grids disconnected from the main flood extent which is an unphysical characteristic of the surrogate. Similar maps for Ishinomaki and Sendai are made available in the Supplement Figures S3 and S4.

We also evaluate the performance of the mean prediction +/- 2 standard deviation using the evaluation metrics for the time series in Table 4. The accuracy score (A) focuses on the identification of the flooded area with depth greater than 10 cm or not, while the goodness of fit (G) provides an assessment of the correct prediction of the maximum inundation depth for the location with depth greater than 10 cm. The general tendency is that misfit reduces when using mean - 2 standard deviation value, highlighting some overestimation in the mean prediction. We plot the distribution of the performance metric G and A for two example events of type A and B for Rikuzentakata using the ensemble of the predictions, to show the how the ensemble captures predictions close to the desired simulation results in Supplement Figures S9 and S10. Across the different test events

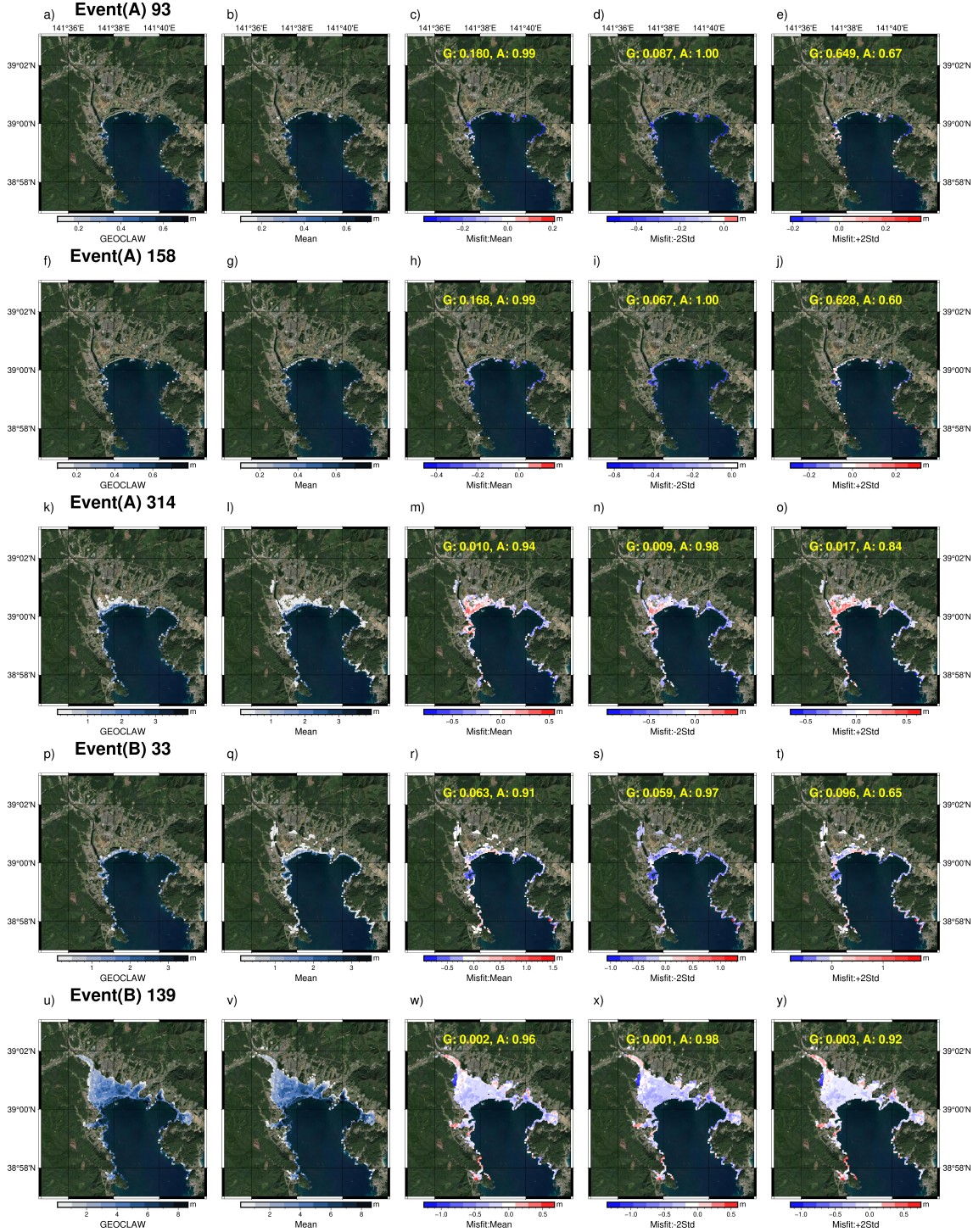

**Figure 14.** Prediction examples from the onshore surrogate at Rikuzentakata for the DOE test events.(Basemap from ESRI World Imagery)

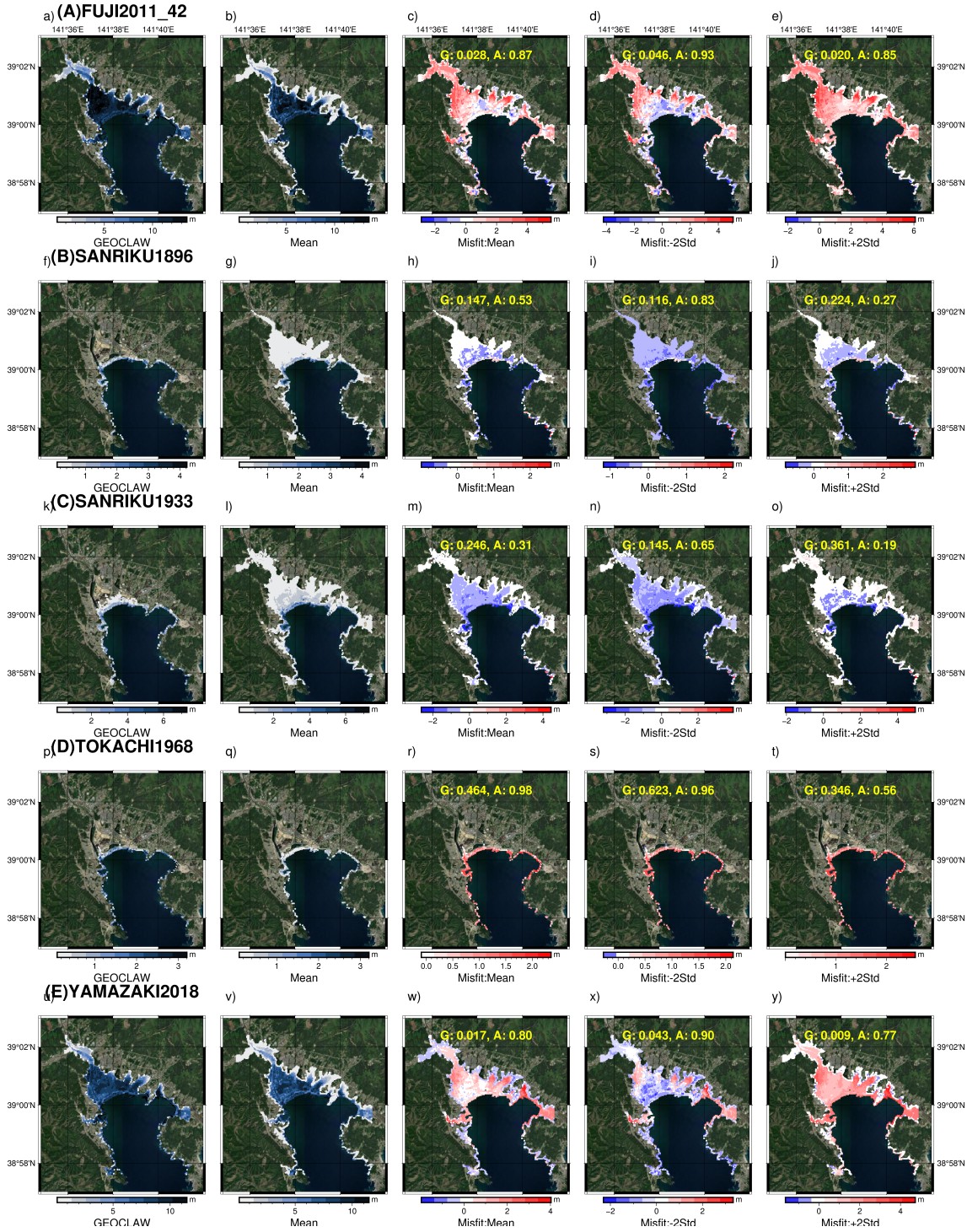

**Figure 15.** Predictions from the onshore surrogate at Rikuzentakata for the historical events.(Basemap from ESRI World Imagery)

and ensemble predictions, 93.397% have an accuracy score greater than 0.9 and G values less than 0.1. The spread of the G and A values for different depth classes are plotted in Fig. 16.

## 4.4 Generalisation Ability : Predicting historical test events

Generalisation of the ML surrogates ensure that they would perform well on unseen data by capturing the underlying characteristics of the data rather than memorising and over fitting on the training data. To test the generalisation we examine the prediction for the historical test events described in 2.1.3. These results are the predictions using all five VED models trained on different data subset in the cross-validation exercise, providing a wider ensemble with 500 predictions compared to the 100 predictions from a single VED model in the previous section.

Figure 13 plots the time series prediction for these events for the three regions, similar to the fold test events in Fig.12. Compared to the results of the fold test events previously described, the peak values and their timing are well captured across the test sites and events. As observed earlier the G values tend to be higher than desirable for the smaller events (B, C and D). The surrogate is unable to predict the high frequency characteristics (event B and C) but is able to capture it in the uncertainty bands of the waveform ensemble. When comparing the mean prediction with the simulation, the G values range between 0.054-0.225 for Rikuzentakata, 0.071-0.442 for Ishinomaki, and 0.07-0.411 for Sendai. Similar to the prediction from the test fold, underestimation in some of the wave peaks result in the MPD of up to 107 min and 65 min in Rikuzentakata and Ishinomaki, respectively, and 193 min in Sendai.

Figure 15 plots the inundation map for Rikuzentakata similar to the fold test events shown in Fig.14. For the large magnitude 2011 Tohoku events (test events A and E) the accuracy metric for the flood mapping is high: 0.9 using the mean prediction for event A and 0.93 for event E. However, there is significant under-prediction in flood depth and misfit up to 5 m. For the smaller magnitude events related to the Sanriku events (test events B and C), there is overestimation in the flood mapping, with many locations having predictions of small depths resulting in lower accuracy metric of 0.83 for event C and 0.65 for event B. For the 1933 Tokachi-Oki event (test event D), accuracy in flood mapping is high with a value of 0.96, but flood depths are underestimated with a low G value of 0.623.

For Ishinomaki the prediction performs well in characterising the small events in terms of the depth and area flooded. There is an underestimation in flood depths and mapping area for event A and an overestimation in flood area for event E. For Sendai the accuracy in the flood mapping footprint is high but significantly underestimated in the prediction of the flood depth for the large magnitude 2011 Tohoku events (test events A and E). Refer to Figures S5, S6,S7 and S8 for the flood prediction and evaluation metrics.

There is high accuracy in mapping of the flood area, but there is a tendency to misrepresent the flood depths in the prediction by the onshore model for the test events. We examine this misfit by plotting the accuracy score (A) for flood mapping and goodness of fit (G) for the different depth classes 0.1–1 m, 1– 2 m, 2– 5 m, 5–10 m, 10–15 m for test events in section 2.1.3 individually and events in 2.1.2 lumped together, at the three test sites as shown in Fig. 16.

To provide an overview of the surrogate model's behaviour, we conducted a comparative analysis between the mean prediction of the test events from the Design of Experiments (DOE) and historical events. In Fig.17(a), we plot the prediction

performance, assessed by G, with the magnitude of the event represented by the L2Norm. Our analysis reveals a degradation in G values for events with L2Norm values below 10. This trend can be attributed to the optimisation process driven by the use of MSE as the loss function, which tends to prioritise accuracy for larger values. Furthermore, in Fig.17(b), we compare the coefficient of determination between predictions and simulation against G as a metric for prediction performance. This visualisation effectively identifies areas where the model performs exceptionally well, but also highlights events where low G value can have a poor COD value due to poor prediction for a portion of the time series, as seen for test event A for Rikuzentakata, or over prediction as seen for the maximum inundation depths for event B in Rikuzentakata.

## 5 Discussion and conclusions

A key challenge in using neural network surrogates for probabilistic tsunami hazard analysis (PTHA) is their limited validation in generalising beyond the specific event types used during training. For instance, their performance on events from different regions, involving multiple sources and mechanisms, has not been sufficiently validated in previous studies. By testing ML surrogates on a diverse set of events, this research broadens their applicability and makes a significant contribution to validating the generalisation capabilities of ML surrogates for PTHA. Furthermore, training datasets often require thousands of events from each source region to fully capture the inherent variability. In many cases, previous studies using ML have prioritised accurate predictions for larger magnitude events, which are crucial for early warning systems. However, for a comprehensive PTHA, it is equally important that surrogates can predict both large and small magnitude events with high accuracy. Recognising this, we designed two specialised ML surrogates to address the distinct challenges of approximating nearshore and onshore tsunami hazards that help offset the related computational costs, while overcoming the limitations of imbalanced and limited training datasets. The nearshore surrogate predicts the time history of tsunami wave at the shore, while the onshore surrogate predicts the inundation depths across vast locations overland. The hybrid ensemble approach introduced here leverages model and parameter sensitivity to enhance prediction accuracy, marking a step forward in integrating uncertainty quantification into ML-based PTHA. This expanded capability, which has been under explored in prior research, is essential for extending tsunami hazard and risk assessment.

The data preparation, model architecture, and training procedures developed here represent key advancements in overcoming the limitations discussed. We train and test our surrogates using a synthetic tsunami dataset modelled using ruptures with both heterogenous and uniform slip. Additionally, the surrogate was also tested for historical events modelled with different rupture models and source locations to assess their effectiveness and investigate the influence of such uncertainties in the initial displacement on surrogate prediction.

To enhance prediction information, we implemented a hybrid ensemble approach that uses multiple variational models. The effectiveness of surrogate models depends on the tuning of latent variables, which vary on the type of prediction parameter, size of output and complexity of the region. Our analysis suggests that more latent variables are necessary for the nearshore surrogate compared to the onshore surrogate i.e., more information needs to be compressed for time series prediction than

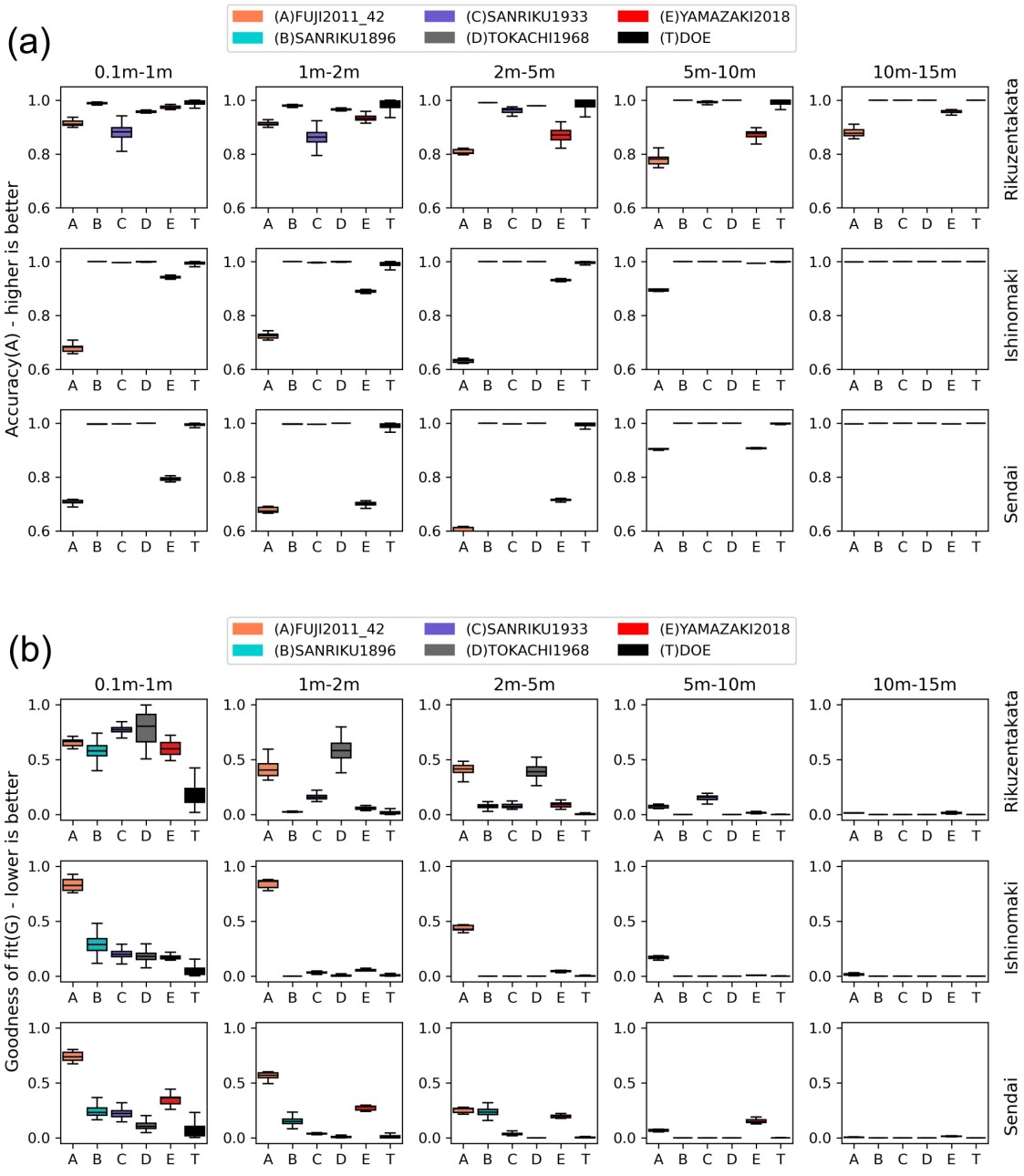

**Figure 16.** Assessing the variability of prediction performance across different depth classes, events, and test sites: (a) Accuracy and (b) Goodness of fit.

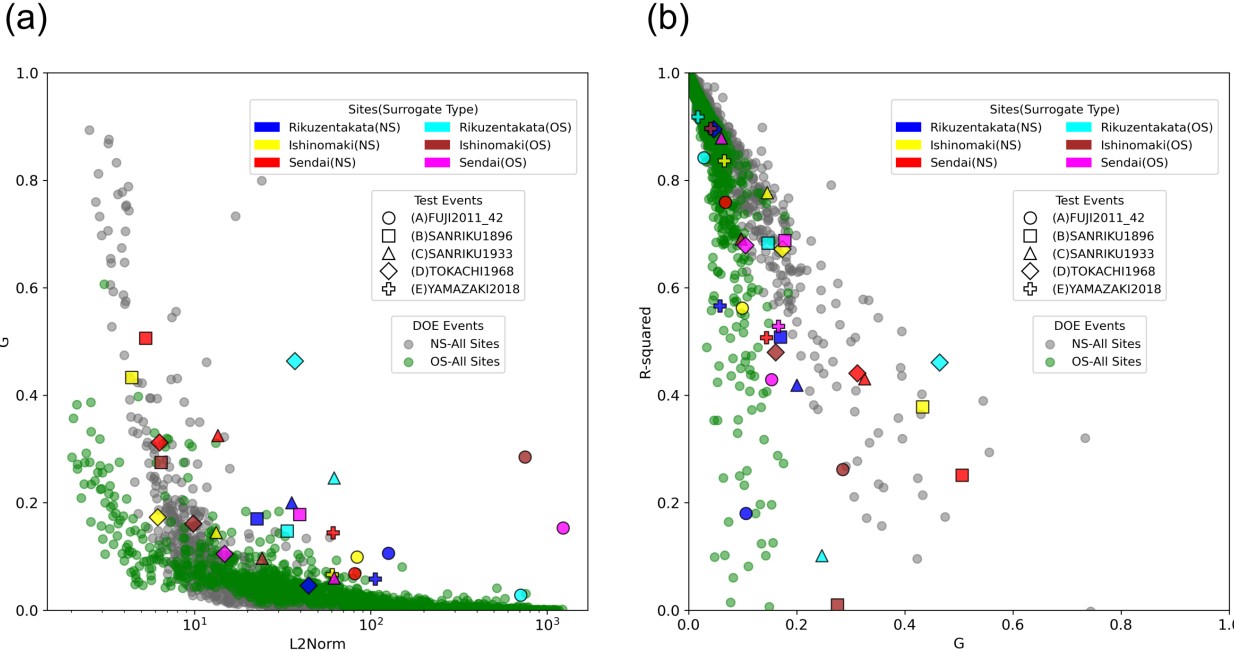

**Figure 17.** Assessing the variability of performance given by G with respect to: (a) the magnitude of the event as indicated by the L2Norm and (b) the coefficient of determination $R^2$

inundation. Also, the complexity of the region has more influence than the size of the prediction locations. Cross-validation
testing confirms the feasibility of using the surrogate in hazard analyses.

The surrogate models exhibit excellent performance in predicting time series data, effectively characterising waveforms for different locations. Despite the peculiar source characteristics and input time series compared to the training dataset, the models accurately capture temporal dynamics of the events. However, high-frequency characteristics are not captured for some test events, which can be due to the choice of shallow or low number of convolution layers and limited information in the training
dataset. There are also certain portions with misrepresented phases which could be due to not including the local deformation explicitly in the learning framework but instead as a pre-processing step.

Inundation prediction, however, presents greater challenges due to the large number of locations and asymmetry in flood occurrence across the prediction locations resulting in artefact or disconnected flooding points and under-prediction in depth in the generalisation testing. Unlike time series prediction with fixed time steps, inundation prediction requires mapping to the
large number of predicted locations (ranging from 6648 to 129941) tested in the study. Addressing these challenges requires further optimisation and refinement of the model architecture, which will be investigated in future studies.

While artefact flooding can be removed with simple post processing routines, there is significant potential for further improvement in the onshore decoder. We set the grid resolution in our inundation simulation at 50 m due to our computational

constraints, using a higher resolution around 5-10 m would mean significantly more points. Implementing multi-head decoders for prediction will provide more efficient and accurate predictions. Other architectural improvements and training methods when using imbalanced datasets, such as weighted sampling, up-sampling of data, and curriculum learning should also be investigated but are beyond the scope of this article. A customised loss function balancing predictions across small and large values by weighting could further enhance model performance and address discrepancies observed in inundation prediction.

Our surrogate model demonstrates stable performance, predicting both small and large events. For we observe good results for events outside the training parameter space. This highlights the versatility of surrogate models in the prediction of events with different magnitudes and characteristics. Although our training data did not consider outer rise mechanisms, the model generalises well as also seen with the 1933 Sanriku Tsunami. This underscores the promise of further improving the method by incorporating a broader range of training data and using better source modelling techniques, as done by (Liu et al., 2021; Núñez et al., 2022). Our analyses show that the nearshore model outperforms the onshore model in terms of generalisation for the historical events.

The training information from our DOE is constrained by both the quality and quantity of events. Recognizing this limitation is crucial for guiding future improvements in the experimental design and training dataset along with advances in the model architecture and training. First, the geographic focus is limited to the Tohoku subduction source region and modelled with a simple scheme, restricting the diversity of the training data and impacting the model's ability to generalise to other regions or varied tsunami scenarios, such as the historic test events (b, c, and d). Second, there is an event imbalance for inundation, particularly for the onshore surrogate, where more events of large inundation are needed to provide sufficient training scenarios at locations far from the coast, which are rarely inundated in the training dataset. Finally, the generalisation test on the onshore surrogate highlights varying prediction accuracies across different test sites, at Rikuzentakata, Sendai, and Ishinomaki. These variations reflect the complexities and limitations of the DOE, where certain test sites with more complex inundation patterns are not well represented in the training data, leading to less accurate predictions.

To improve our understanding of ML models in tsunami prediction, further efforts are needed in acquiring more extensive training and testing datasets, conducting benchmarking and comparison studies with other surrogates or test regions and making them available open source. In summary, our study demonstrates the potential of surrogate models in accurately predicting tsunami hazard variables, while also highlighting areas for further refinement to improve model performance and reliability in hazard assessment tasks.

**Table 6.** Runtime for tsunami numerical simulation using GeoClaw

| Total Cell Updates | Device type | Parallelisation | Time taken per event(hr) | No of events | Total compute time(hr) |
|---|---|---|---|---|---|
| $0.531 \times 10^{11}$ | CPU | 10 CPU threads | 3.45 | 564 | 1945 |

Understanding the impact of the size of the training dataset and the variability is important and should be examined. Topography and coastal morphology, the number of model inputs, type of prediction, and the tsunami sources considered among other factors, all contribute to the size of training data and compute resource needed, and the resulting surrogate prediction

**Table 7.** Runtime for machine learning training for the different region specific surrogates

| | Region | Max epoch | Time taken per training(sec) | No of folds | Total training time(min) |
|---|---|---|---|---|---|
| **Nearshore Surrogate** | Rikuzentakata | 3000 | 54 | 5 | 4.5 |
| | Ishinomaki | 3000 | 72 | 5 | 6 |
| | Sendai | 4000 | 73 | 5 | 6 |
| **Onshore Surrogate** | Rikuzentakata | 20000 | 156 | 5 | 13 |
| | Ishinomaki | 20000 | 167 | 5 | 14 |
| | Sendai | 20000 | 177 | 5 | 15 |
| | | | | **Total** | 58.5(1 hr) |

performance. The run time information for the tsunami numerical simulation using a CPU device Intel Xeon Silver 4216 CPU with 2.1 Ghz, 313 GB RAM is provided in Table 6 and for the ML training from this work in Table 7 using GPU device NVIDIA A100 80GB.

The computation time for training an ML surrogate is in minutes, this results in a remarkable efficiency gain of approximately 2000 times during the model inference when applying the ML surrogate, as compared to running the full simulation. We should also consider that multiple training runs are needed for fine-tuning the model architecture, hyperparameters and other choices at hand for the ML model which was not quantified here. Standardising the model architecture and procedures along with the use of pre-trained models for transfer learning can help minimise these training related costs.

The nearshore approximation approach is useful as a dynamic hazard proxy at the coast where considering the arrival time of tsunami and its peaks is important for studying applications such as evacuation planning. However, the direct application may be limited by the need to train the model for each location. This may hinder the application at scale, but the option to predict the inundation maps helps to efficiently assess large regions. Our approach can help extend offshore tsunami hazard information (Davies et al., 2018; Basili et al., 2021) available at deep offshore points to the much-needed onshore hazard and risk, with a relatively limited number of simulations and associated computational costs. The learning from this study can easily be adapted for other intensity measures or parameters like velocity or momentum flux, but also finds use in early warning settings where direct tsunami sensors are not yet available. Similar computational challenges also exist for other coastal flood hazards like storm surge, coastal flooding due to riverine, tropical cyclone rainfall and compound flood drivers, and use of such surrogate can also be investigated in these cases using different set of input parameters.

*Code availability.* The tsunami simulations use GeoClaw, Pytorch was used as the ML framework in python, other analysis was conducted using bespoke scripts in python, maps were prepared using Pygmt package in python and QGIS. All scripts and codes used in the project and those for the preparation of the paper are available at https://github.com/naveenragur/tsunami-surrogates.git

*Data availability.* The Japan Cabinet Office Data from the modelling study by special Investigation Committee on Subduction-type Earthquakes around the Japan Trench and Chishima Trench (Central Disaster Prevention Council) are publicly available at https://www.geospatial. jp/ckan/organization/naikakufu-002. The COP-DEM (https://doi.org/10.5069/G9028PQB) are available through OpenTopography from https://portal.opentopography.org/raster?opentopoID=OTSDEM.032021.4326.3.The observed water level, astronomical tide level, and tidal level deviation observed from the 2011 off the Pacific coast of Tohoku earthquake tsunami is publicly available from NOWPHAS at https:

535 //nowphas.mlit.go.jp/pastdata. The 2011 off the Pacific coast of Tohoku earthquake tsunami information - field survey results used for the validation of the GeoClaw model are publicly available from the Coastal Engineering Committee of the Japan Society of Civil Engineers website at https://www.coastal.jp/tsunami2011. The fault parameters used to model the initial displacement for the historic events are publicly available at the links cited in the references. Large input files for GeoClaw and their post-processed outputs used to train and test the surrogate models are available at https://doi.org/10.5281/zenodo.10817116

*Author contributions.* N.R.R., K.J., M.P. and M.M conceived and designed the research. N.R.R conceived the experiments, carried out the simulation, developed the ML model and performed the training. All authors analysed, reviewed the results and wrote the manuscript.

*Competing interests.* The contact author has declared that none of the authors has any competing interests

*Acknowledgements.* N.R.R.was supported by the research scholarship of the project "Dipartimenti di Eccellenza", funded by the Italian Ministry of Education, University and Research at IUSS Pavia.

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

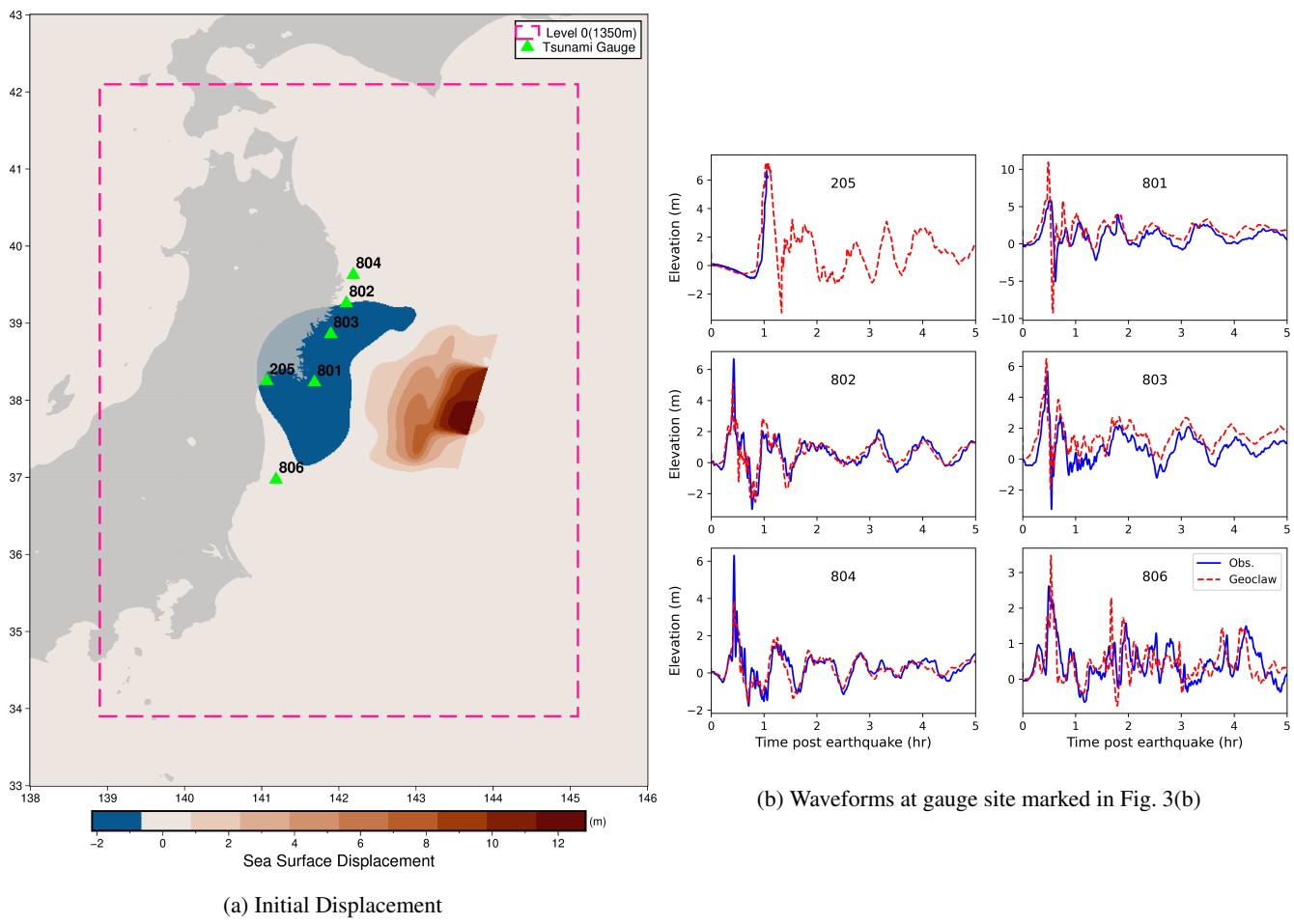

(a) Initial Displacement

(b) Waveforms at gauge site marked in Fig. 3(b)

**Figure S1.** Plots of simulated tsunami inundation using source (Fujii et al., 2011) compared to the actual observation for the 2011 Tohoku tsunami event

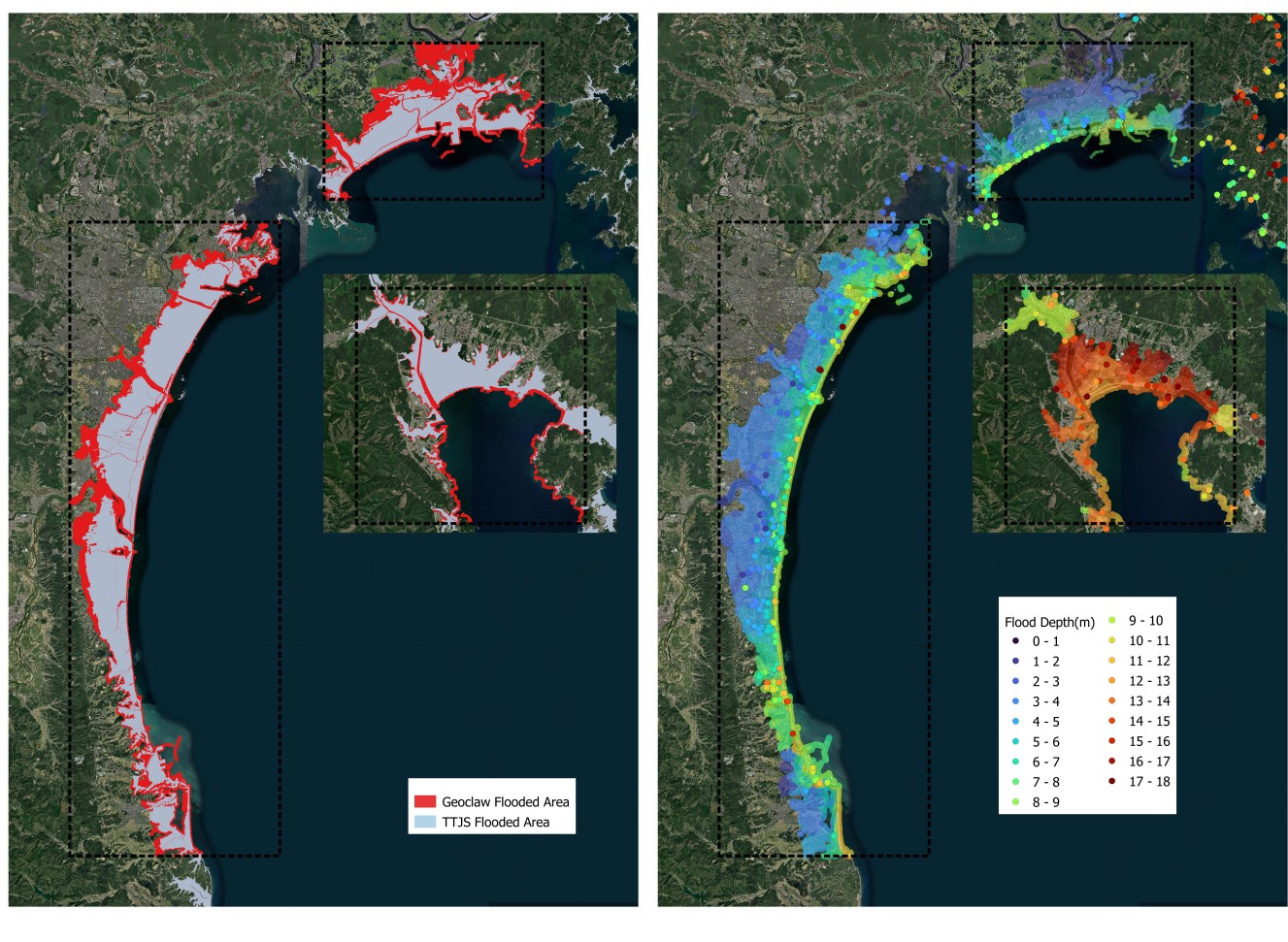

(a) Flooded Area

(b) Flooded Depth

**Figure S2.** Plots of simulated tsunami inundation using source (Fujii et al., 2011) compared to the actual observation for the 2011 Tohoku tsunami event.(Basemap from ESRI World Imagery)

## Surrogate architecture and configuration

**Table S1.** Model Parameters of the VED network - Nearshore

| Layer Number | Layer Type | Input Channels or Size | Output Channels or Size | Activation | Pooling Operation |
|---|---|---|---|---|---|
| **Offshore Encoder** | | | | | |
| 1 | Conv1d | input | 64 | Leaky ReLU (0.5) | MaxPool1d (2x2) |
| 2 | Conv1d | 64 | 64 | Leaky ReLU (0.5) | MaxPool1d (2x2) |
| 3 | Conv1d | 64 | 128 | Leaky ReLU (0.5) | MaxPool1d (2x2) |
| 4 | Conv1d | 128 | 128 | Leaky ReLU (0.5) | MaxPool1d (2x2) |
| 5 | Linear | [8192] | [2 x Zdim] | | |
| **Variational Encoding** | | | | | |
| 6 | Reparametrise | [2 x Zdim] | [Zdim] | | |
| **Nearshore Decoder** | | | | | |
| 7 | Linear | [Zdim] | [128 x 64] | | |
| 8 | ConvTranspose1d | 128 | 128 | Leaky ReLU (0.5) | MaxPool1d (2x2) |
| 9 | ConvTranspose1d | 128 | 64 | Leaky ReLU (0.5) | MaxPool1d (2x2) |
| 10 | ConvTranspose1d | 64 | 64 | Leaky ReLU (0.5) | MaxPool1d (2x2) |
| 11 | ConvTranspose1d | 64 | output | Leaky ReLU (0.5) | MaxPool1d (2x2) |

**Table S2.** Model Parameters of the VED network - Onshore

| Layer Number | Layer Type | Input Channels [Size] | Output Channels [Size] | Activation | Pooling Operation | Other Operations |
|---|---|---|---|---|---|---|
| | | | **Offshore Encoder** | | | |
| 1 | Conv1d | input | 64 | Leaky ReLU (0.5) | MaxPool1d (4x4) | BatchNorm1d |
| 2 | Conv1d | 64 | 64 | Leaky ReLU (0.5) | MaxPool1d (4x4) | |
| 3 | Conv1d | 64 | 128 | Leaky ReLU (0.5) | MaxPool1d (4x4) | |
| 4 | Conv1d | 128 | 128 | Leaky ReLU (0.5) | MaxPool1d (4x4) | Dropout (0.1) |
| 5 | Linear | [512] | [2 x Zdim] | | | |
| | | | **Variational Encoding** | | | |
| 6 | Reparametrise | [2 x Zdim] | [Zdim] | | | |
| | | | **Onshore Decoder** | | | |
| 7 | Linear | [Zdim] | [64] | | | |
| 8 | Linear | [64] | output | | | |

**Performance metrics for the generalisation test using historic events**

**Table S3.** Model performance statistics for different events at various sites using the mean prediction from the nearshore surrogate.

| Site | Event | G | $R^2$ | MSE | L2Norm |
|---|---|---|---|---|---|
| Rikuzentakata | FUJI2011_42 | 0.106 | 0.18 | 10.196 | 125.919 |
| | SANRIKU1896 | 0.17 | 0.508 | 0.246 | 22.612 |
| | SANRIKU1933 | 0.2 | 0.419 | 0.717 | 35.587 |
| | TOKACHI1968 | 0.046 | 0.895 | 0.201 | 44.266 |
| | YAMAZAKI2018 | 0.058 | 0.566 | 4.044 | 106.158 |
| Ishinomaki | FUJI2011_42 | 0.099 | 0.562 | 1.727 | 83.648 |
| | SANRIKU1896 | 0.433 | 0.378 | 0.012 | 4.404 |
| | SANRIKU1933 | 0.145 | 0.777 | 0.037 | 13.236 |
| | TOKACHI1968 | 0.173 | 0.672 | 0.012 | 6.189 |
| | YAMAZAKI2018 | 0.066 | 0.836 | 0.533 | 60.553 |
| Sendai | FUJI2011_42 | 0.068 | 0.759 | 1.395 | 80.902 |
| | SANRIKU1896 | 0.506 | 0.251 | 0.02 | 5.292 |
| | SANRIKU1933 | 0.325 | 0.431 | 0.1 | 13.566 |
| | TOKACHI1968 | 0.312 | 0.441 | 0.022 | 6.322 |
| | YAMAZAKI2018 | 0.144 | 0.507 | 1.724 | 61.162 |

**Table S4.** Model performance statistics for different events at various sites using the mean prediction from the onshore surrogate.

| Site | Event | G | $R^2$ | MSE | L2Norm |
|---|---|---|---|---|---|
| Rikuzentakata | FUJI2011_42 | 0.028 | 0.842 | 3.774 | 706.438 |
| | SANRIKU1896 | 0.147 | 0.683 | 0.051 | 33.559 |
| | SANRIKU1933 | 0.246 | 0.102 | 0.474 | 61.982 |
| | TOKACHI1968 | 0.464 | 0.461 | 0.105 | 37.078 |
| | YAMAZAKI2018 | 0.017 | 0.918 | 1.213 | 511.5 |
| Ishinomaki | FUJI2011_42 | 0.285 | 0.262 | 3.712 | 747.957 |
| | SANRIKU1896 | 0.275 | -0.684 | 0.001 | 6.475 |
| | SANRIKU1933 | 0.097 | 0.691 | 0.003 | 24.125 |
| | TOKACHI1968 | 0.161 | 0.48 | 0.001 | 9.82 |
| | YAMAZAKI2018 | 0.041 | 0.896 | 0.158 | 337.745 |
| Sendai | FUJI2011_42 | 0.153 | 0.429 | 2.575 | 1233.79 |
| | SANRIKU1896 | 0.178 | 0.688 | 0.004 | 39.573 |
| | SANRIKU1933 | 0.06 | 0.879 | 0.004 | 61.973 |
| | TOKACHI1968 | 0.105 | 0.679 | 0.001 | 14.827 |
| | YAMAZAKI2018 | 0.166 | 0.528 | 2.256 | 1108.09 |

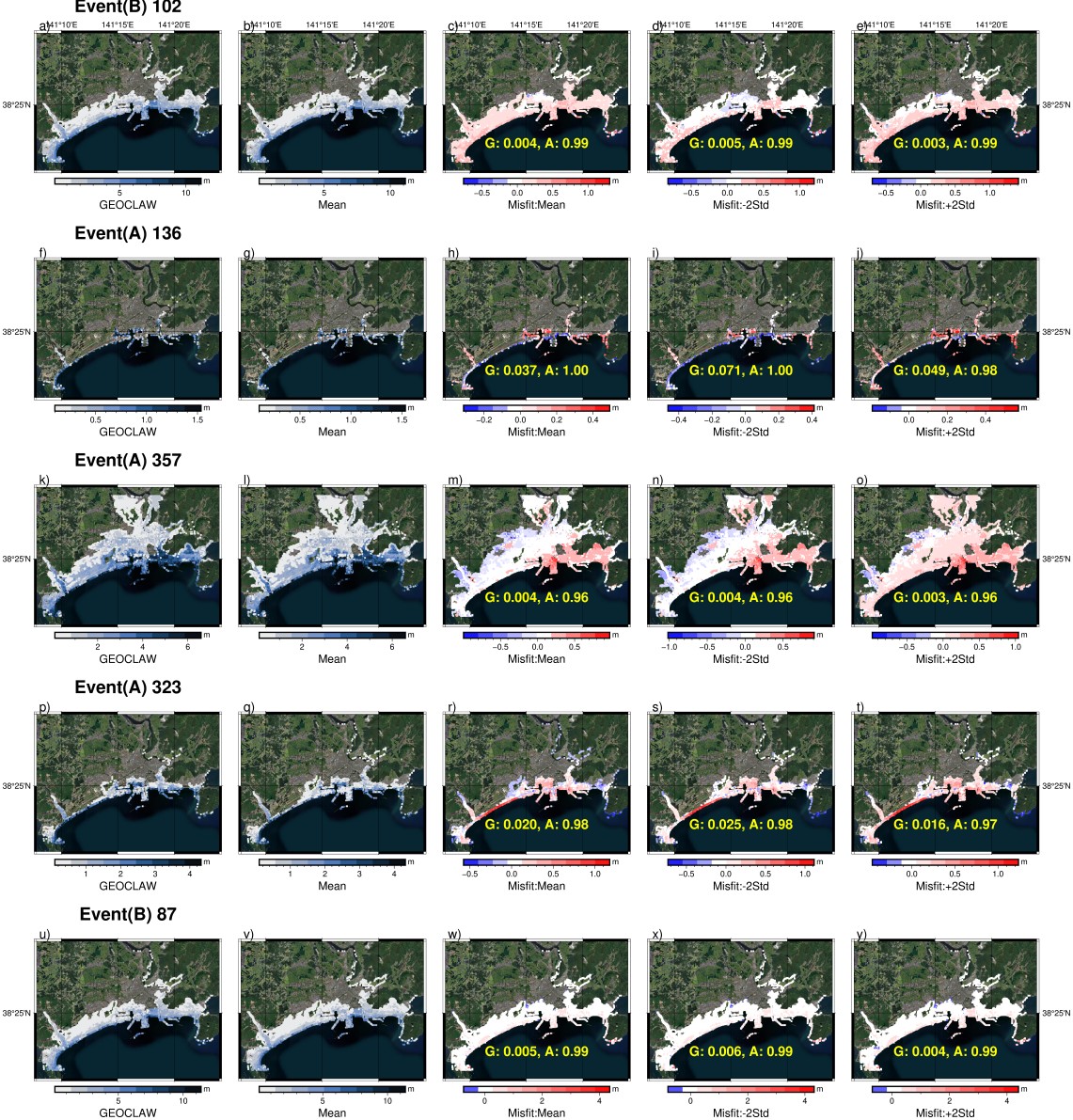

**Figure S3.** Prediction examples from the onshore surrogate at Ishinomaki for the test events.(Basemap from ESRI World Imagery)

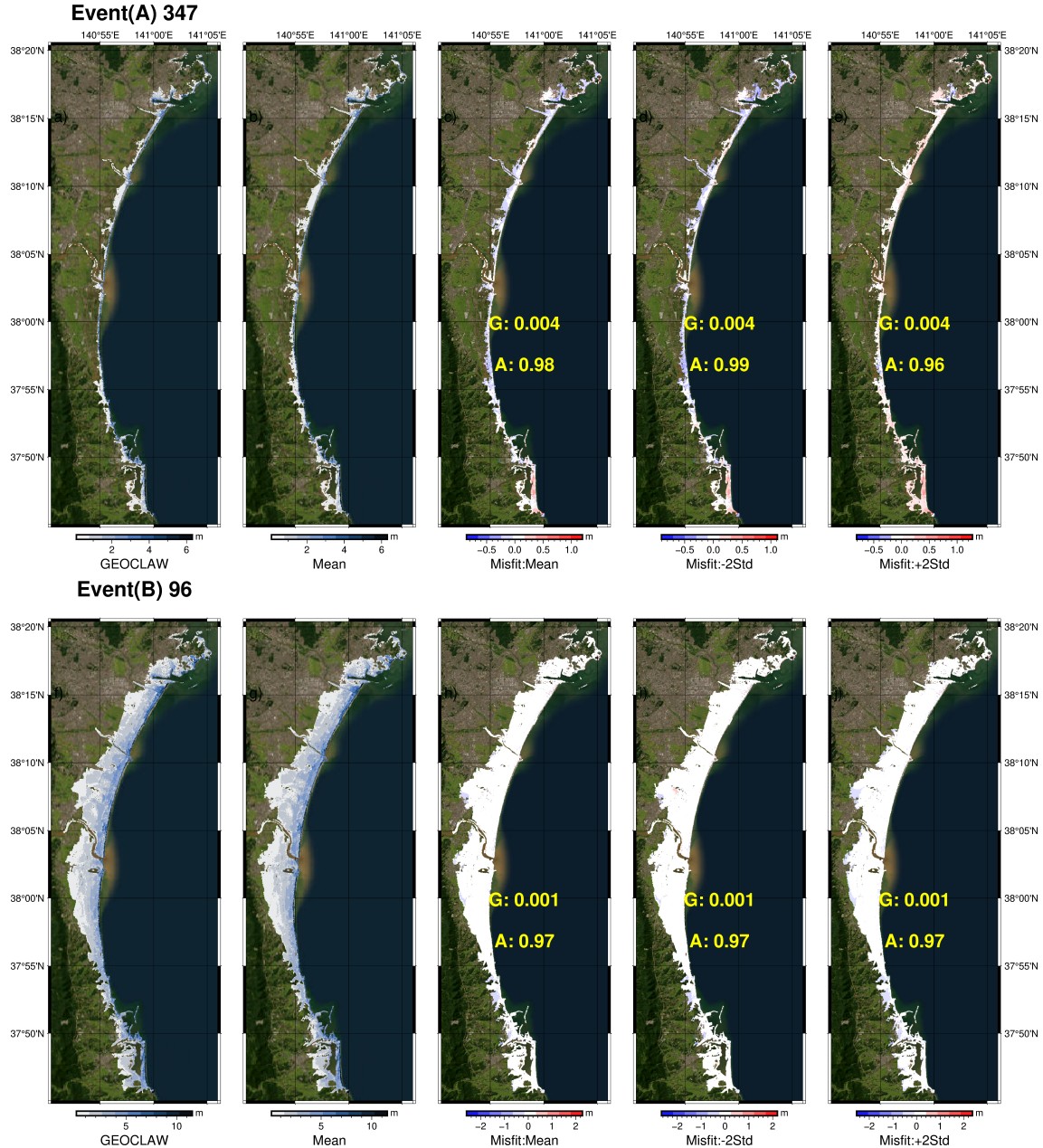

**Figure S4.** Prediction examples from the onshore surrogate at Sendai for the test events.(Basemap from ESRI World Imagery)

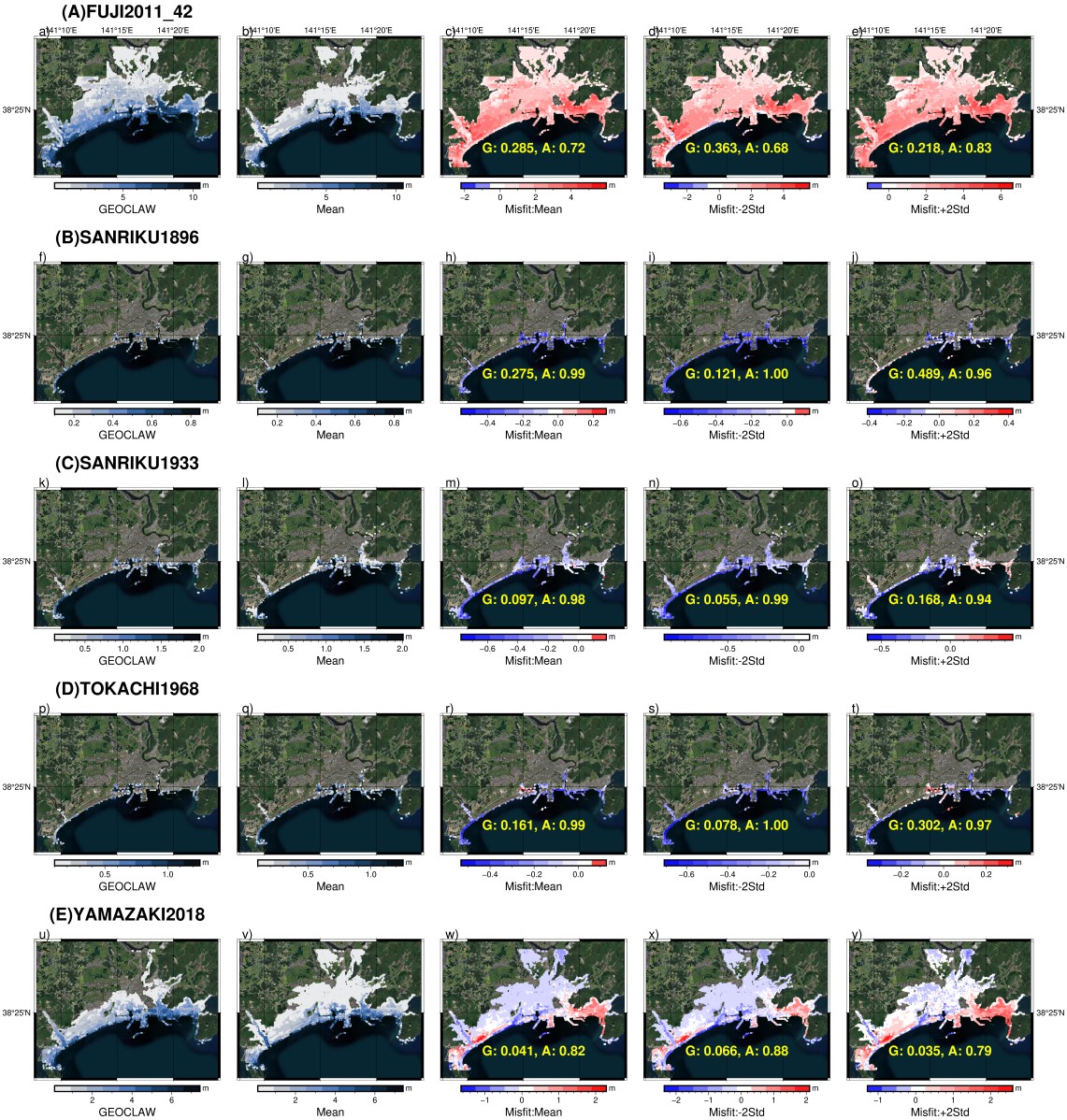

**Figure S5.** Historical prediction from the onshore surrogate at Ishinomaki.(Basemap from ESRI World Imagery)

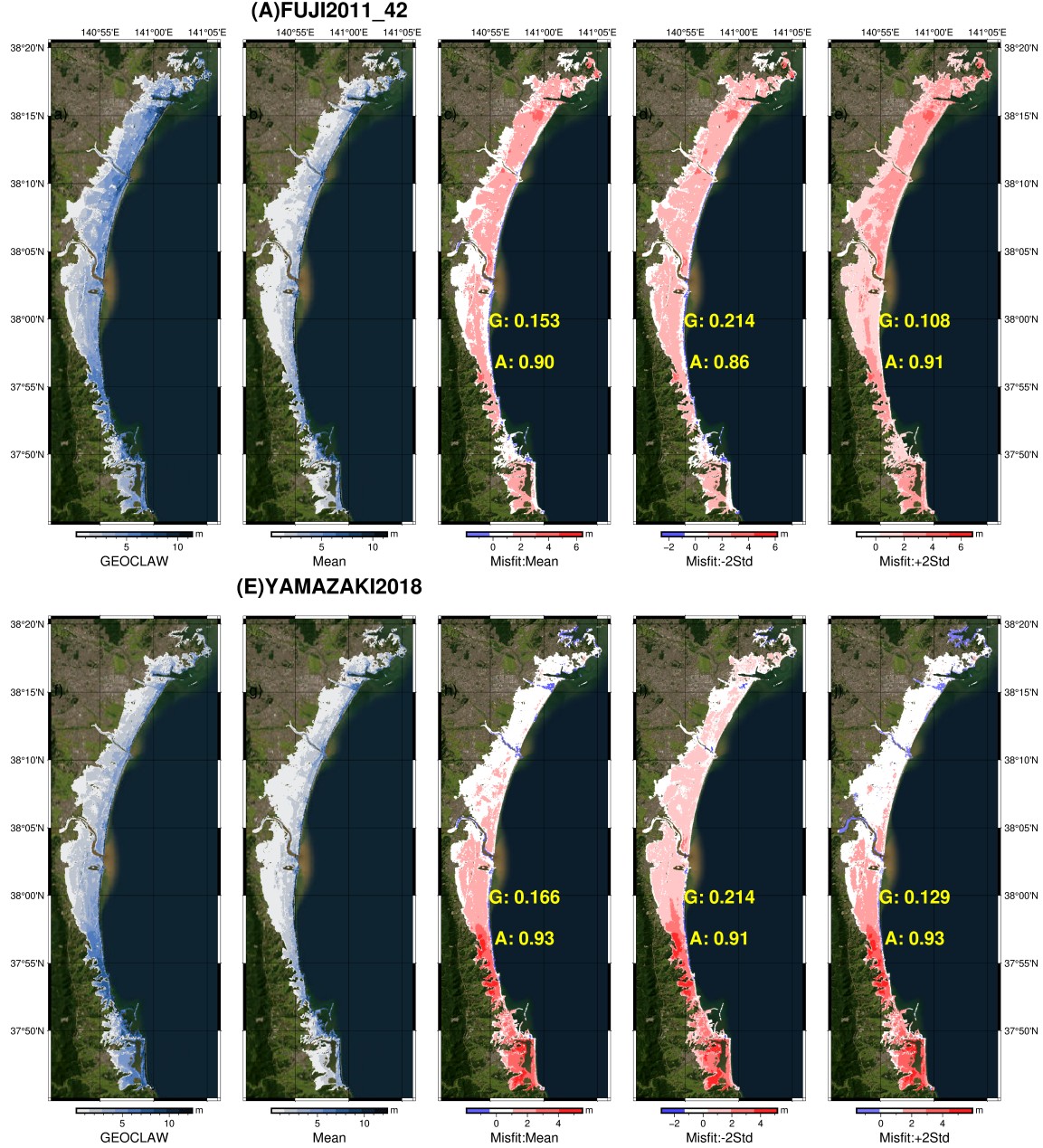

**Figure S6.** Historical prediction for 2011 Tohoku events from the onshore surrogate at Sendai.(Basemap from ESRI World Imagery)

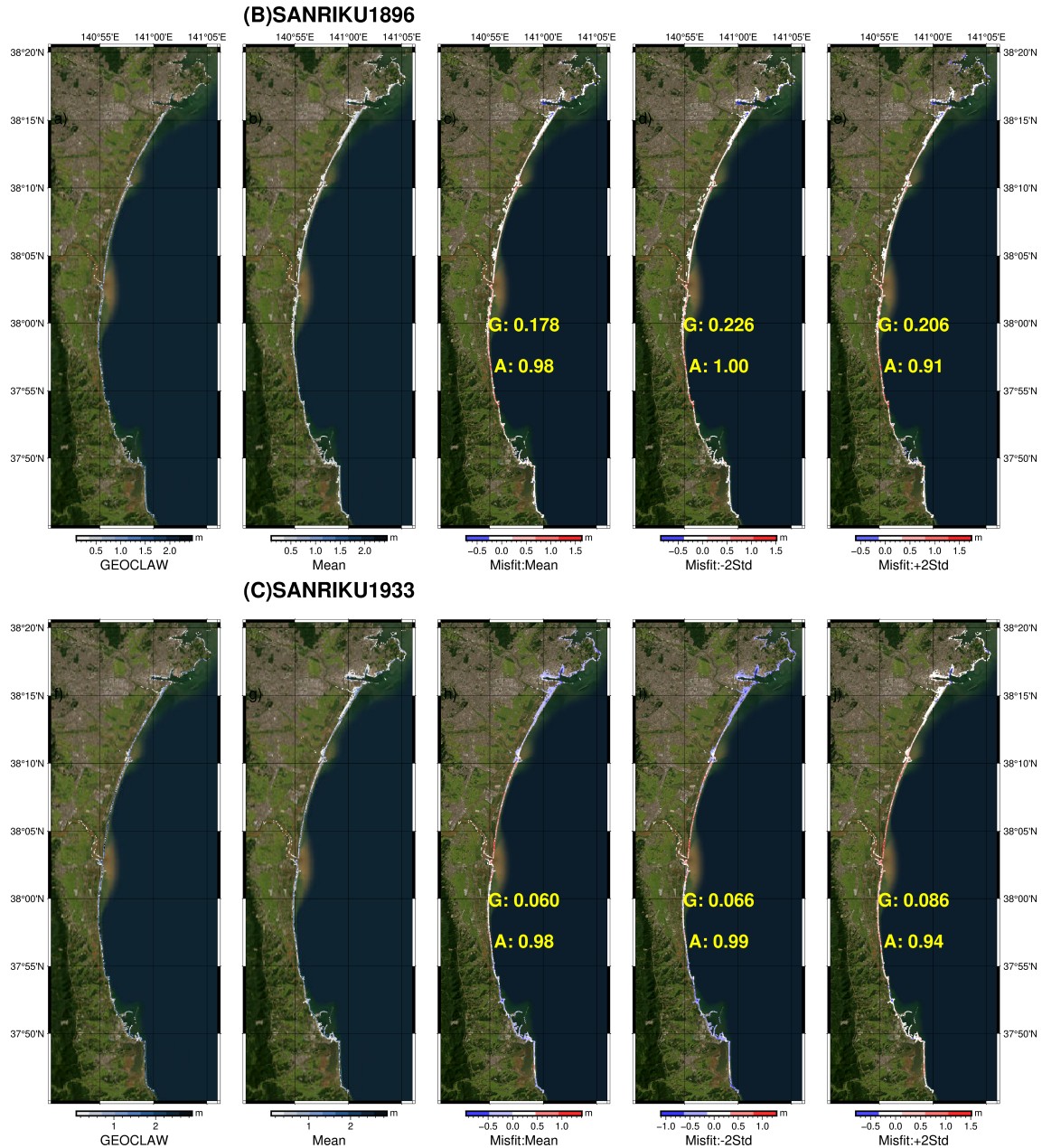

**Figure S7.** Historical prediction for the Sanriku events from the onshore surrogate at Sendai.(Basemap from ESRI World Imagery)

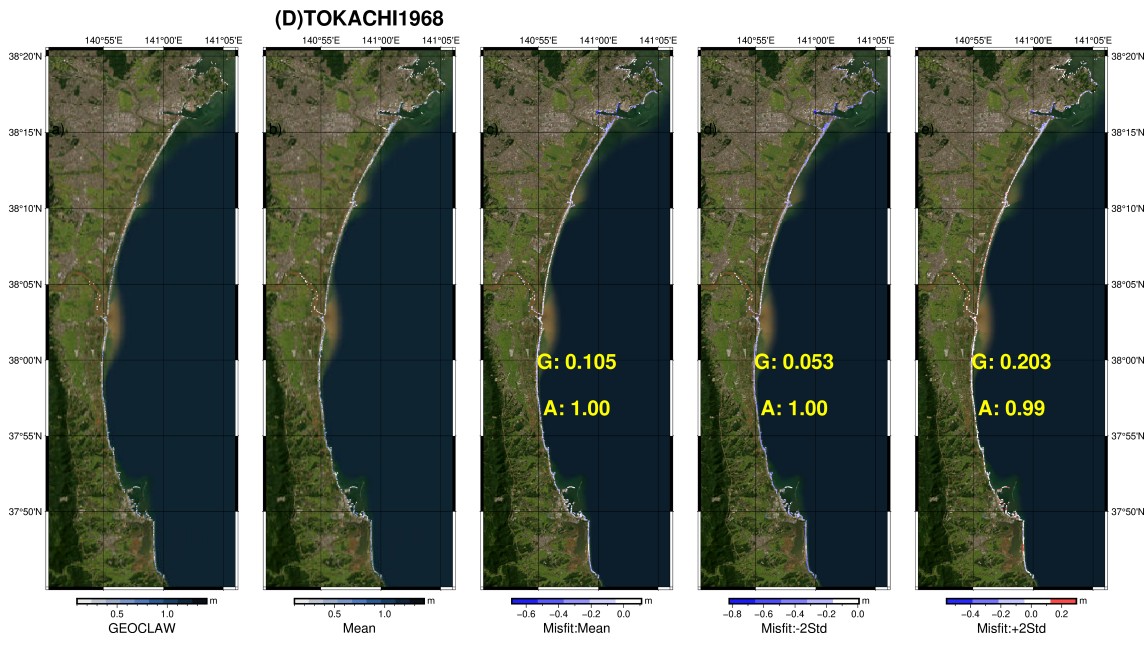

**Figure S8.** Historical prediction for the Tokachi-Oki event from the onshore surrogate at Sendai.(Basemap from ESRI World Imagery)

## Benefit of ensemble predictions

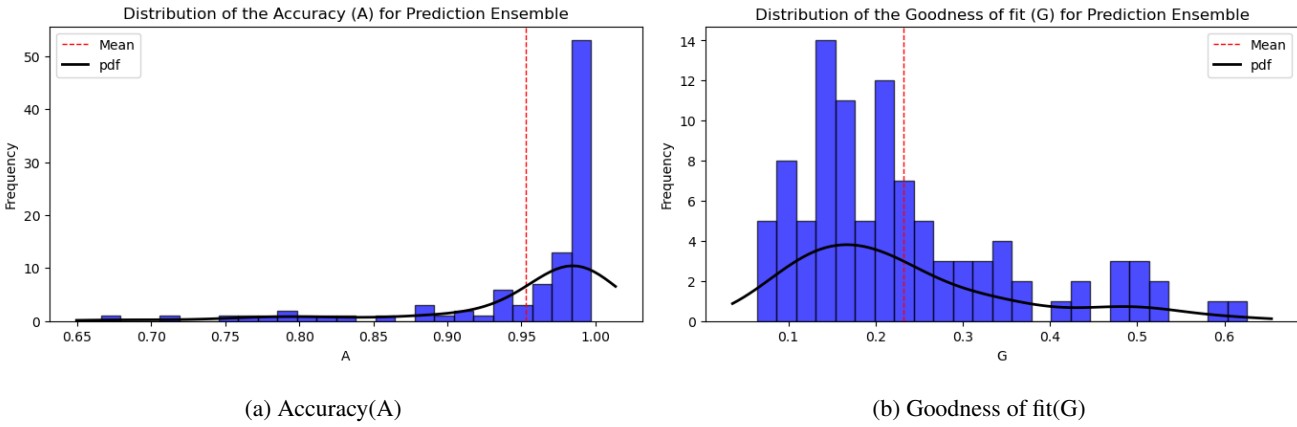

(a) Accuracy(A)  (b) Goodness of fit(G)

**Figure S9.** Assessing the variability of the performance for the prediction ensemble for the test event 158 of type A at Rikuzentakata

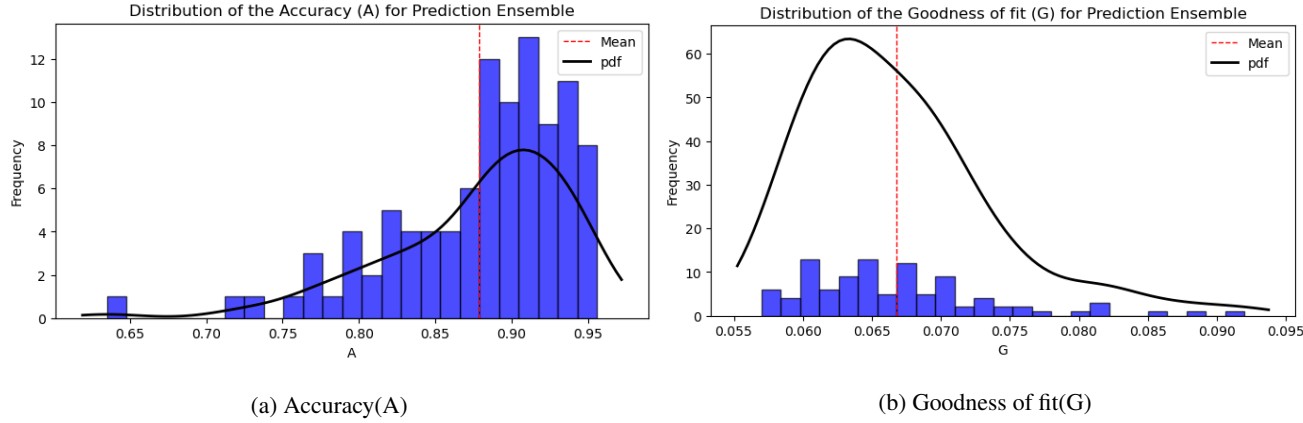

(a) Accuracy(A)  (b) Goodness of fit(G)

**Figure S10.** Assessing the variability of the performance for the prediction ensemble for the test event 33 of type B at Rikuzentakata