# Peer review of "Advancing nearshore and onshore tsunami hazard approximation with machine learning surrogates"

_Natural Hazards and Earth System Sciences, 2024_

## Author Comment (AC1)

**Responses by authors in blue:**

We thank the Referee#1 for their valuable comments and suggestions which improves the quality of the manuscript. Detailed responses are provided to your questions. Blue text shows our response, updates which will be incorporated in the manuscript are highlighted in red as track change and black text shows the referee comments.

The paper presents an advance in tsunami hazard approximation, offering a machine learning (ML)-based solution to the computational challenges of traditional methods. However, for publication, major revisions are necessary to address the following concerns.

**Details on Test Sets**

The paper does not provide sufficient details on the features of the test sets used for model validation. Specific characteristics and parameters of the test sets should be clearly stated to assess the model's generalizability. Additionally, the rationale behind selecting specific test sets is only briefly mentioned. A more detailed justification for their selection is necessary. Include a detailed description of the test sets, including their characteristics. Explain how these sets represent the diversity of potential tsunami scenarios and justify their choice to enhance the study's credibility.

**Response:**

We assume that the reference here is to the test sets used from the design of experiments (DOE) during the k-fold cross-validation (CV) study. K-fold CV is widely used for ML model evaluation, especially when dealing with limited data, as it ensures that each data point is used for both training and validation, enhancing the robustness of the evaluation. In our study, we utilize a 5-fold CV approach, where the dataset is randomly partitioned into five equal-sized folds. Each fold is used once as the test set while the remaining four folds are used for model training. This process is repeated five times, with each fold serving as the test set exactly once. The characteristics and parameter ranges for each test fold as when all the test folds are combined, mirror those of the overall training dataset. This is summarized in Table 2 and Figure 6.

To address the reviewer's request for specific details, we updated the following information into the revised manuscript:

Updated Figure 6. shared below to also describe the earthquake magnitude in the training set for the three regions, along with outliers.

[Figure]

**In section 2.2.2, Line 296 is modified as:**

*"Given the relatively small size of our training dataset, we employ K-fold cross-validation with five folds to fully use the available data set for training and testing. The data set is randomly partitioned into five equal-sized folds. Each fold is used once as the test set while the remaining four folds are used for model training. This process is repeated five times, with each fold serving as the test set exactly once. The characteristics and parameter ranges for each test fold as when all the test folds are combined together mirror that of the overall training dataset as found in Table 2 and Figure 6. This helps in assessing the model's generalisation ability and sensitivity to overfit with such limited training information (Mulia et al., 2020)."*

*Reference: Mulia, I. E., Gusman, A. R., and Satake, K.: Applying a Deep Learning Algorithm to Tsunami Inundation Database of Megathrust Earthquakes, Journal of Geophysical Research: Solid Earth, 125, e2020JB019 690, https://doi.org/10.1029/2020JB019690, eprint:https://onlinelibrary.wiley.com/doi/pdf/10.1029/2020JB019690, 2020.*

**Model Training and Weights**

The paper does not discuss whether different weights were assigned to various types of events during training. Assigning different weights could help ensure the model does not overfit to more frequent, less severe events, thus improving its predictive performance for rare, high-impact events. Elaborate on the training process, including whether different weights were assigned to events and how this impacted model performance. If not used, consider implementing and discussing the potential benefits of such weighting schemes.

**Response:**  In traditional classification problems, approaches such as over sampling, under sampling, and reweighting are commonly used to mitigate class imbalance problems. However, for our tsunami surrogate models, the definition of imbalance is more complex and multifaceted. It can be based on various factors, including earthquake characteristics (e.g., magnitude and hypocentre location), waveform input characteristics (e.g., maximum amplitude), or output characteristics (e.g., maximum inundation depths and coverage).

Understanding and addressing the influence of such imbalances in the training set is crucial. While assigning weights to different types of events is a common approach, it requires a detailed understanding of how these weights would impact the sensitivity and performance of the model. This is a subject of our forthcoming work, where we will investigate these aspects in detail using evaluation on a large simulation database.

In the current study, we did not explicitly assign different weights to various events. Instead, we focused on the robust design of the surrogate model and implemented several regularization techniques to handle the potential overfitting and inherent imbalance. We employ shallow neural network layers, with regularization schemes such as max-pooling operations, dropout, and variational latent spaces to ensure the model can scale appropriately for different magnitudes of events and their associated input and output parameters. Our model's performance, particularly for higher water levels in time series and water depths in inundation, shows stability and significant skill above an L2Norm value of 10, as illustrated in Figure 17(a). Additionally, Figures 16(a) and 16(b) demonstrate that the accuracy (A) and goodness of fit (G) metrics tend to improve for higher depth classes in both DOE test sets and historical events. This also indicates that the model does not overfit to more frequent, less severe events.

Furthermore, in the initial stages of our work, we explored other approaches to mitigate imbalance or scaling issues, which were not all beneficial:

1. **Scaling and normalizing** the dataset based on training data.
2. **Curriculum learning scheme**, gradually train the model from lower magnitude (Mw) events to higher ones.
3. **Alternative output parameters**, considered using max inundation height as the output parameter for the onshore surrogate.
4. **Supplement the training dataset** with more events(type B), initially our dataset only consisted of type A events.

In conclusion, while we did not use explicit weighting schemes in the current study, we acknowledge the potential benefits of such approaches. Our future work will delve deeper into understanding the impact of event weighting and other balancing schemes to enhance model performance and generalizability as suggested in Section.5 L441.

**Information Leakage**

The paper lacks details on measures taken to prevent information leakage, which can lead to overly optimistic performance estimates if the test data inadvertently influences the training process. Clearly outline the steps taken to ensure strict separation between training and test data. Discuss any data augmentation techniques and how they are managed to avoid information leakage. This

transparency will strengthen the reliability of the reported results. What do you think about the possibility of information leakage for the 2011 Tohoku test case?

**Response:** To prevent information leakage, but also overfitting our study adhered to the following protocol:

1. **No data augmentation**: We did not use data augmentation techniques in our study.
2. **Data splitting and cross-validation**: As discussed in our response to the comment above on "Details on Test Sets," we employed a 5-fold cross-validation (CV) approach. In this method, the dataset is randomly partitioned into five equal-sized folds. Each fold is used once as the test set, while the remaining four folds are used for training. This process ensures that every data point is used for both training and validation, and it helps prevent any inadvertent information leakage that could occur with a single train-test split.
3. **Ensemble of variational encoder-decoder (VED) Models**: To mitigate the risk of overfitting, sensitivity to parts of the training data, information leakage, we used an ensemble of models for the generalisation tests on historical events. Each model in the ensemble was trained on different subsets of the training data derived from the 5-fold CV process. The final predictions were generated by aggregating the results from these models. This ensemble approach helps capture the overall uncertainty due to variability in the training dataset and the stochastic nature of model parameterization in the latent space, enhancing the robustness and reliability of the results.

This ensures that there is no contamination of the training data by the test data, providing an unbiased evaluation of the model's generalization capabilities for the historic events like 2011 Tohoku test case.

**Conclusions**

The results shown in Figure 15 suggest that the quality of training sets is more important than the quantity. For the test cases outside the design of experiments, there are clear discrepancies between the prediction and observation, which is an intrinsic nature of ML algorithms. Include a discussion on the design of experiments, including the limitations. This will provide a more comprehensive understanding of the model's strengths and areas needing improvement.

**Response:** That is an important remark, that designing a training dataset limited in size but with sufficient variability is vital for training such tsunami surrogate. We will add an additional paragraph to the discussion section as follows:

*"The training information from our DOE is constrained by both the quality and quantity of events, recognizing its limitations is crucial for guiding future improvements in the experimental design and training dataset along with advances in the model architecture and training. First, the geographic focus is limited to the Tohoku subduction source region and modelled with a simple scheme, restricting the diversity of the training data and impacting the model's ability to generalize to other regions or varied tsunami scenarios, such as the historic test events (b, c, and d). Second, there is an event imbalance for inundation, particularly for the onshore surrogate, where more events of large inundation are needed to provide sufficient training scenarios at locations far from the coast, which are rarely inundated in the training dataset. Finally, the generalization test on the onshore surrogate highlights varying prediction accuracies across different test sites, at Rikuzentakata, Sendai, and Ishinomaki. These variations reflect the complexities and limitations of the DOE, where certain test sites with more complex inundation patterns are not well-represented in the training data, leading to less accurate predictions."*

**Minor comments:**

L15 Tsunami -> **Tsunamis**

L16 USD 280 billion damage -> USD 280 billion **in** damage

L29 a past historical event -> historical events

L91 machine learning-based -> ML-based; You may replace machine learning with ML in other places of the paper.

L193 **Inconsistency in labeling the test events.** Make sure the labels are consistent including Test A&E in the 2011 Tohoku case, Figure 5 & 15.

L458 remove "it contains"

**Response:**

Thank you for your observations, listed correction will be updated in the revised manuscript.

---

## Author Comment (AC2)

**Responses by authors in blue:**

We thank the Referee#2 for their valuable comments and suggestions which improves the quality of the manuscript.  Detailed responses are provided to your questions. Blue text shows our response, underline updates which will be incorporated in the revised manuscript are highlighted as track change and black text shows the referee comments.

The study uses a surrogate approach based on a variational encoder-decoder (VED) to predict the tsunami time series at different depths and maximum inundation depths at three coastal sites in Japan. The surrogate accuracy is validated against historical rupture scenarios. I add some comments below that I believe could strengthen the work presented.

Comments:

The design of experiments is not very clear in terms of number of scenarios and input variables. The authors mention 559 events split to 383 and 176 depending on the nature of the rupture. More information is needed on how these numbers were selected, and also the parameter ranges that led to the variation in magnitudes (length, width, displacement). Furthermore, more clarification is needed on whether there are any other input variables beside the moment magnitude (e.g. location of the event) that are varied in the surrogate development.

**Response:**

Thank you for highlighting the need for a more detailed description of the design of the experiment (DOE). We updated the section providing more details as reads below.

[revised manuscript text omitted]

I would suggest adding an outline of the times 1) for building the two ML surrogates, 2) for prediction and 3) to run the deterministic model. Possibly in the form of a matrix, this should showcase the benefits of using a surrogate approach.

**Response:** This is provided as the supplement Tables S5 and S6; we briefly touch on this at line 60 and 460. In the revised manuscript we will move this to the section 5. (Discussion and conclusions) and present it as part of the discussion.

In 310 and elsewhere in the manuscript please replace observations/observed with model/modelled or simulations/simulated as it can be confused with physical observations of the event.

**Response:** Thanks for suggesting this, we will update this in the revised manuscript.

In table 5 there seems to be a lot of variance regardless the number of the fold. In some cases, increasing the fold reduces the SME but in other cases it increases the SME. Is this variance random or based on certain conditions?

**Response:** Thank you for your observation. We would like to clarify that we conducted k-fold cross-validation with 5 folds (k=5) exclusively. Each column in Table 5 represents the results from the evaluation using the withheld test set for each fold iteration. Consequently, each column corresponds to a unique combination of random events used for training and testing, with no repetition across fold iterations. The primary purpose of calculating the Mean Squared Error (MSE) across different folds is to assess the model's sensitivity, identify poor fits, and detect signs of overfitting for that specific portion of the training dataset. The variance in MSE observed across folds can indeed reflect differences in the performance of the surrogate model depending on the subset of data used for training. We will update the text to include the below explanations:

For the nearshore surrogate, the relatively consistent MSE across folds suggests that the model is robust, with no significant overfitting and an overall good fit to the data. However, for the onshore surrogate, there is noticeable sensitivity to the training data, as evidenced by slightly higher MSE values for certain locations, such as Rikuzentakata (fold 2) and Ishinomaki (fold 0). This increased sensitivity could be attributed to two factors, namely the training size and the complexity of the output. The smaller size of the training set may lead to higher variance in model performance, as the model has less data to learn from, making it more sensitive to the specific events included in each training fold. The onshore surrogate is tasked with predicting inundation, which is inherently more complex and variable compared to nearshore waveforms. This complexity can further lead to greater variation in model performance across different folds, especially with limited training data.

In summary, the observed variance in MSE across folds is not random but is influenced by the size and complexity of the training data. For the nearshore surrogate, the model demonstrates stable performance, while the onshore surrogate shows some sensitivity, which is expected given the challenging nature of the inundation predictions.

The legends in the figures should be more descriptive, especially in figures 10, 11 the red, blue symbols, lines and black dotted line. In these figures I would assume that these are simulated outputs instead of observed? In a similar manner for figures 12 and 13 for dotted lines, uncertainty bounds etc.

**Response:** Yes, you are correct. We will update the figure with better legend to prescribe that, this is a comparison between the values simulated by GEOCLAW numerical simulation and the prediction from surrogate along with its uncertainty bounds.

As below:

[Figure]

In figure 12, the predictions of events 14, 139, 102, 87 and 96 match nearly identically to the simulations with very small uncertainty bounds. Are those events in close proximity to other events in the training dataset?

**Response:** Yes, there are events in close proximity in the training events occurring at same location occur but with a different slip and magnitude. Our surrogate model also has a tendency to fit better for events with larger values as function the loss function built with the MSE component which penalises larger misfit more (also noticeable in the prediction results in Fig.14 available in the next comment).

When considering, for example event 14 of type B for Rikuzentakata. The events in proximity from the training set are events where the same 6 faults rupture but with a different slip distribution. Event 12 is the closest match we found shown in the figures below.

[Figure]

In figure 14, the misfit between predictions and simulations and +-2 standard deviations and simulations (columns 3,4 and 5) seem to be very close in terms of values, possibly because the standard deviations are small? Can the authors provide an example with values and how including the standard deviation reduces the misfit between prediction and simulation?

**Response:**

Thank you for this question regarding the Fig.14. We have updated the events selection to present more diverse examples; see updated Figures below. In the prediction examples from the DOE test set here, the relatively small standard deviation reflects the high accuracy and low variance of the surrogate's prediction. This variance reduces especially as the events get bigger in magnitude. To illustrate how including the standard deviation affects the misfit and accuracy we provide the measure of G and A as an indicator along with the the examples for the mean, +- 2 standard deviation Fig.14. Here higher accuracy (A values close to 1) represents good prediction of the inundation extent while good fit (G close to 0) represents good prediction in the inundation depth. The general tendency is that misfit reduces when using mean-2 standard deviation, highlighting some overestimation in the mean prediction.

[Figure]

We additionally plot the distribution of the performance metric G and A for two events using the ensemble of the predictions, to show the how the ensemble captures predictions close to the desired simulation results.

**Event 158(Type A)**

[Figure]

[Figure]

**Event 33(Type B)**

---

## Referee Report (RR1)

Review on the manuscript "Advancing Nearshore and Onshore Tsunami Hazard Approximation with Machine Learning Surrogates"

This manuscript introduces an approach in tsunami hazard and risk assessment through the use of machine learning (ML) surrogates. While the manuscript has been clearly improved since earlier versions, further revisions are required before it can be considered for publication.

**Major comments:**

While the manuscript is well-organized overall, some sections, particularly "Discussion and Conclusions", are overly detailed and repetitive. For instance, the discussion often reiterates technical results already covered in previous sections.

L437 "The general challenge with the use of such neural network surrogates for use in PTHA is that they have not been comprehensively validated if they generalise well beyond the specific type of events on which they are trained, such as events with different source mechanisms or when considering tsunami generated from multiple source regions for PTHA."
This sentence was unclear to me. I am not sure what you are trying to convey, and you may my suggestion if this looks fine:
"A challenge of using neural network surrogates in PTHA is their limited validation for generalizing beyond the specific event types used during training. For instance, their performance on events with different regions or multiple sources lacks sufficient validation."

L441 "The emphasis is on accurate predictions for larger magnitudes, which is crucial for early warning purposes. However, for the surrogate to serve as an effective hazard approximation in PTHA, it must accurately represent both large and small magnitude events despite limited training data."
This part also needs to be rewritten. The authors are trying to discuss the imbalance of the training data due to the significance of large events. However the meaning of the following sentence "it must accurately …" is unclear.

L444 Remove "extend upon previous work and"
L448 "in a general setting and the influence …" This is redundancy.
L485 "events, recognizing" ->"events. Recognizing"
L506 "compute time" -> "computation time" ?

The application and methodology appear to be similar to prior works, and it is unclear what is the significant expansion in this study. It is necessary to clearly state in the conclusion how this study represents an advancement over previous studies.

**Comments:**
Consider using a more specific description. For example, in abstract, the authors use "at large regional scale" and ""large portions of the coast". This expression may be unclear to the readers.

L28 "select"->"limited"

Each subfigure in Figure 16 uses a different y-axis range, which makes it difficult to compare results. Consider using the same y-axis range or stating why the ranges differ.

---

## Author Response (AR2)

**Responses by authors in blue:**

We thank the referee and editor for taking their time to provide valuable comments and suggestions to further revise the manuscript. Here, we provide responses to their questions and comments. Blue text shows our response to the referee comments, updates which will be incorporated in the revised manuscript are highlighted as track change and black text shows the referee comments.

**Review by Referee**

Review on the manuscript "Advancing Nearshore and Onshore Tsunami Hazard Approximation with Machine Learning Surrogates" This manuscript introduces an approach in tsunami hazard and risk assessment through the use of machine learning (ML) surrogates. While the manuscript has been clearly improved since earlier versions, further revisions are required before it can be considered for publication.

We find that summarizing the main point of each test set is very useful for framing the discussion; however, we appreciate the point that we include too many technical details in the "discussion and conclusions" section. We tried to reduce the technical details to broad statements in order to make the discussion more easily readable in the revised manuscript.

**Major comments:**

While the manuscript is well-organized overall, some sections, particularly "Discussion and Conclusions", are overly detailed and repetitive. For instance, the discussion often reiterates technical results already covered in previous sections.

L437 "The general challenge with the use of such neural network surrogates for use in PTHA is that they have not been comprehensively validated if they generalise well beyond the specific type of events on which they are trained, such as events with different source mechanisms or when considering tsunami generated from multiple source regions for PTHA."

This sentence was unclear to me. I am not sure what you are trying to convey, and you may my suggestion if this looks fine:

"A challenge of using neural network surrogates in PTHA is their limited validation for generalizing beyond the specific event types used during training. For instance, their performance on events with different regions or multiple sources lacks sufficient validation."

This has been updated in the revised first paragraph of "Discussion and Conclusions" and reads as below.

A key challenge in using neural network surrogates for probabilistic tsunami hazard analysis (PTHA) is their limited validation in generalising beyond the specific event types used during training. For instance, their performance on events from different regions, involving multiple sources and mechanisms, has not been sufficiently validated in previous studies. By testing ML surrogates on a diverse set of events, this research broadens their applicability and makes a significant contribution to validating the generalisation capabilities of ML surrogates for PTHA. Furthermore, training datasets often require thousands of events from each source region to fully capture the inherent variability. In many cases, previous studies using ML have prioritised accurate predictions for larger magnitude events, which are crucial for early warning systems. However, for a comprehensive PTHA, it is equally important that surrogates can predict both large and small magnitude events with high accuracy. Recognising this, we designed two specialised ML surrogates to address the distinct challenges of approximating nearshore and onshore tsunami hazards that help offset the related computational costs, while overcoming the limitations of imbalanced and limited training datasets. The nearshore surrogate predicts the time history of tsunami wave at the shore, while the onshore surrogate predicts the inundation depths across vast locations overland. The hybrid ensemble approach introduced here leverages model and parameter sensitivity to enhance prediction accuracy, marking a step forward in integrating uncertainty quantification into MLbased PTHA. This expanded capability, which has been under explored in prior research, is essential for extending tsunami hazard and risk assessment.

L441 "The emphasis is on accurate predictions for larger magnitudes, which is crucial for early warning purposes. However, for the surrogate to serve as an effective hazard approximation in PTHA, it must accurately represent both large and small magnitude events despite limited training data."

This part also needs to be rewritten. The authors are trying to discuss the imbalance of the training data due to the significance of large events. However the meaning of the following sentence "it must accurately ..." is unclear.

This has been updated in the revised paragraph above.

L444 Remove "extend upon previous work and" Removed.

L448 "in a general setting and the influence ..." This is redundancy. L485 "events, recognizing" ->"events. Recognizing" L506 "compute time" -> "computation time"? Corrected.

The application and methodology appear to be similar to prior works, and it is unclear what is the significant expansion in this study. It is necessary to clearly state in the conclusion how this study represents an advancement over previous studies.

The updated paragraph above summaries the main advancements in the study, all addressing the applicability of ML surrogates for PTHA and PTRA

- Validation and generalisation testing for broader applicability needed in PTHA.
- Handling imbalanced and limited training information overcoming the challenges of sparse training data.
- Modelling two tsunami hazard measure wave time history and max inundation.
- Uncertainty quantification from the surrogates.

**Comments:**

Consider using a more specific description. For example, in abstract, the authors use "at large regional scale" and "large portions of the coast". This expression may be unclear to the readers.

**We have updated the abstract to clearly mention it now reads as below:**

Probabilistic tsunami hazard and risk assessment (PTHA and PTRA) are vital methodologies for computing tsunami risk and prompt measures to mitigate impacts. However, their application across extensive coastlines, spanning hundreds to thousands of kilometres, is limited by the computational costs of numerically intensive simulations. These simulations often require advanced computational resources like high-performance computing (HPC), and may vet necessitate reductions in resolution, fewer modelled scenarios, or use of simpler approximation schemes. To address these challenges, it is crucial to develop concepts and algorithms for reducing the number of events simulated and more efficiently approximate the needed simulation results. The case study presented herein, for a coastal region of Tohoku, Japan, utilises a limited number of tsunami simulations from submarine earthquakes along the subduction interface to build a wave propagation and inundation database. These simulation results are fit using a machine learning (ML) based variational encoder-decoder model. The ML model serves as a surrogate, predicting the tsunami waveform on the coast and the maximum inundation depths onshore at the different test sites. The performance of the surrogate models was assessed using a five-fold cross-validation assessment across the simulation events. Further, to understand their real world performance and generalisability, we benchmarked the ML surrogates against five distinct tsunami source models from the literature for historic events. Our results found the ML surrogate capable of approximating tsunami hazard on the coast and overland, using limited inputs at deep offshore locations and showcase their potential in efficient PTHA and PTRA.

L28 "select"->"limited"

**Thanks, corrected.**

Each subfigure in Figure 16 uses a different y-axis range, which makes it difficult to compare results. Consider using the same y-axis range or stating why the ranges differ.

Thanks for suggesting this, updated the figures as below: